EMBO
reports

# Pseudo-repeats in doublecortin make distinct mechanistic contributions to microtubule regulation

Szymon W Manka[**] & Carolyn A Moores[*]

## Abstract

**Doublecortin (DCX) is a neuronal microtubule-associated protein (MAP) indispensable for brain development. Its flexibly linked doublecortin (DC) domains—NDC and CDC—mediate microtubule (MT) nucleation and stabilization, but it is unclear how. Using high-resolution time-resolved cryo-EM, we mapped NDC and CDC interactions with tubulin at different MT polymerization stages and studied their functional effects on MT dynamics using TIRF microscopy. Although coupled, each DC repeat within DCX appears to have a distinct role in MT nucleation and stabilization: CDC is a conformationally plastic module that appears to facilitate MT nucleation and stabilize tubulin–tubulin contacts in the nascent MT lattice, while NDC appears to be favored along the mature lattice, providing MT stabilization. Our structures of MT-bound DC domains also explain in unprecedented detail the DCX mutation-related brain defects observed in the clinic. This modular composition of DCX reflects a common design principle among MAPs where pseudo-repeats of tubulin/MT binding elements chaperone or stabilize distinct conformational transitions to regulate distinct stages of MT dynamic instability.**

**Keywords** cryo-EM; DCX domain; doublecortin; microtubule-associated protein; pseudo-repeat

**Subject Categories** Cell Adhesion, Polarity & Cytoskeleton; Structural Biology

## Introduction

The highly dynamic and regulated network of microtubules (MTs) in all eukaryotic cells plays numerous critical roles throughout development. MT-associated proteins (MAPs) spatially and temporally control the MT cytoskeleton (Atherton *et al*, 2013), thereby governing cell morphology, polarity, intracellular organization, and cell division. MAPs are thus central to organismal morphogenesis and maturation.

Doublecortin (DCX) is the founding member of the family of MAPs that makes significant contributions to MT regulation in metazoan development (Gönczy *et al*, 2001; Bechstedt *et al*, 2010; Fourniol *et al*, 2013). In humans, the *Dcx* gene is located on chromosome X. It is indispensable for brain development and specifically for migration of immature neurons, such that lack of functional DCX manifests in gray matter heterotopia (females) or lissencephaly (smooth brain; males), causing various degrees of intellectual disability and epilepsy (Gleeson *et al*, 1998; des Portes *et al*, 1998). Although DCX is predominantly expressed in developing brain, it is also a marker of adult neurogenesis (Moreno-Jiménez *et al*, 2019), and in several cancers, where its abnormal expression may promote metastasis by facilitating cell migration (Ayanlaja *et al*, 2017).

Microtubules are built of α/β-tubulin heterodimers that assemble longitudinally into protofilaments (PFs) and laterally into hollow cylinders. *In vitro*, MTs can form with various PF numbers, but in mammalian cells they mainly contain 13 PFs (Tilney *et al*, 1973; Chaaban & Brouhard, 2017). DCX strongly promotes nucleation and stabilization of this physiological 13-PF architecture (Moores *et al*, 2004). The 13-PF lattice is pseudo-helical, with 12 PFs connected by homotypic (α-α and β-β) lateral contacts and one site, called the seam, built from heterotypic (α-β and β-α) contacts. The molecular mechanisms by which MTs are nucleated, grow with specific architectures and shrink, have been extensively studied but are still not completely understood.

Like numerous other MAPs, DCX is built from pseudo-repeats of MT/tubulin-binding elements. It contains two homologous globular doublecortin (DC) domains—NDC and CDC—connected by a 40-residue unstructured linker (Kim *et al*, 2003; Cierpicki *et al*, 2006), followed by a disordered C-terminal serine/proline-rich domain (Fig 1A–C, Appendix Figs S1 and S2). NDC and CDC share a similar ubiquitin-like fold but differ in amino acid sequence (24% sequence identity; Fig 1B). Together, the DC domains have been implicated in MT binding and many disease-causing mutations cluster within them (Sapir *et al*, 2000; Taylor *et al*, 2000; Bahi-Buisson *et al*, 2013; Appendix Fig S2A), but

Institute of Structural and Molecular Biology, Department of Biological Sciences, Birkbeck, University of London, London, UK
*Corresponding author. Tel: +44 (0) 20 3926 3516; E-mail: c.moores@mail.cryst.bbk.ac.uk
**Corresponding author. Tel: +44 (0) 20 7679 5147; E-mail: s.manka@mail.cryst.bbk.ac.uk

whether the sequence differences between them reflect distinct contributions to DCX function is less clear. Previous structural studies of individual domains have suggested that the biological properties of NDC and CDC are distinct (Kim *et al*, 2003; Cierpicki *et al*, 2006; Burger *et al*, 2016; Rufer *et al*, 2018). However, isolated DC domains alone do not stimulate MT polymerization, i.e., a DC domain tandem is required for this activity (Taylor *et al*, 2000), and a distinct contribution of each DC domain to MT lattice-based function of DCX has been indicated (Bechstedt & Brouhard, 2012; Liu *et al*, 2012; Bechstedt *et al*, 2014; Burger *et al*, 2016), although the mechanism by which this occurs is also not known.

Doublecortin does not bind to unpolymerized tubulin, but stabilizes existing and/or nascent MT assemblies (Moores *et al*, 2006; Bechstedt *et al*, 2014). It recognizes a corner between four tubulin dimers via a DC domain, except at the seam (Fourniol *et al*, 2010; Fig 1A). Cryo-EM studies of DCX-stabilized MTs only reveal density corresponding to a single-DC domain—theoretically this could correspond to NDC, CDC or a mixture of both, and so far, available data have not allowed differentiation between these possibilities (Fig 1A, Appendix Fig S2B).

We hypothesized that each DC domain in DCX makes a distinct contribution to MT nucleation and stabilization, and that the study of DCX could shed light more generally on the mechanisms of these MT processes. We addressed these ideas using biochemistry, TIRF microscopy, and high-resolution time-resolved single-particle cryo-EM. Structural insight into the wild-type DCX (WT) and a chimeric variant, where CDC is replaced with a second NDC (NN), combined with MT dynamics data, uncovered differential roles of the two DC domains. CDC appears to be primarily implicated in MT nucleation, where it also acts to drive specification of 13-PF MTs, whereas NDC appears to collaborate with CDC along the mature MT lattice, providing durable stabilization of MTs.

# Results

## CDC in DCX accelerates MT nucleation and induces 13-protofilament MT architecture

To dissect the roles of the individual DC domains in MT nucleation and stabilization we began by preparing a series of single and double-DC domain constructs (Fig 1C, Appendix Figs S1C and S2C). The purity of all constructs was verified by SDS–PAGE and their intrinsic thermal stability was tested by thermal shift assay (Appendix Fig S3). WT, NN, and all the single-DC domain constructs had melting temperatures ($T_m$) above 60 °C. The construct with swapped DC domains (CN) produced a less sharp thermal shift, but the apparent $T_m$ was ~60 °C (Appendix Fig S3C). In contrast, the CDC-CDC (CC) construct had the lowest apparent thermal stability with $T_m$ of approximately 40 °C (Appendix Fig S3C), suggesting that it is likely partially unfolded at 37 °C, consistent with earlier NMR data (Kim *et al*, 2003).

We measured bulk MT nucleation and polymerization in the presence of our constructs at 37 °C using a turbidity assay, combined with verification of tubulin polymerization products by negative stain and cryo-electron microscopy (EM; Fig 1C–F, Appendix Fig S4). We found that only WT and NN efficiently nucleated MTs at 5 μM tubulin concentration—below the ~10 μM critical concentration (Walker *et al*, 1991)—and that this activity was dose-dependent (Fig 1C–F, Appendix Fig S4). MTs sporadically occurred at higher concentrations (2 μM) of CN, but products in this reaction were predominantly protein aggregates (Appendix Fig S4B). CC induced only protein aggregation (Appendix Fig S4B). Thus, both CN and CC were considered unstable and were not analyzed further. None of the single-DC domain constructs up to concentrations equimolar to tubulin (5 μM) nucleated MTs, even with both DC domains added *in trans* (Fig 1C, Appendix Fig S4A), suggesting that the physical connection between the DC domains is important. The

---

**Figure 1. Each DC domain in DCX contributes a distinct function in regulation of MT assembly.**

A   Two routes of MT polymerization in the presence of DCX are considered: direct nucleation, involving stabilization of early tubulin oligomers by DCX; stabilization, involving DCX binding only between straight PFs of a complete MT lattice. Question marks, unknown mode of DCX binding to tubulin/MT.

B   Human DCX schematic, sequence alignment of the DC domains from human isoform 2 (Uniprot ID: O43602-2) with indication of secondary structures and 3D structure alignment, in which selected amino acid positions are indicated using single-letter codes. Sequence identity is 24.1% by Clustal 2.1 (Larkin *et al*, 2007). H, α-helix; S, β-strand.

C   List of DCX constructs with their abbreviated names and a summary of their activity. Construct boundaries are indicated with residue numbers. +, WT activity; +/−, impaired activity compared to WT; ++, enhanced activity compared to WT; −, no activity; nd, not determined.

D   Plot of initial MT polymerization velocities at 37 °C in the presence of increasing concentrations of WT or NN. Tubulin concentration = 5 μM. Each point represents an average of three independent measurements of the rate of turbidity increase measured at 500 nm wavelength (OD500/min). Fitted are sigmoidal dose-response curves: $Y = Y_B + ((Y_T − Y_B)/(1 + 10^{Xi − X}))$, where $Y_B$ and $Y_T$ are Y values (OD500/min) at the bottom and top plateaus, respectively, and $Xi$ is the X value at the inflection point. Representative MT samples from 2:5 μM WT:tubulin or NN:tubulin incubations are shown in (E) and (F), respectively. MT architecture distributions for WT and NN are shown with pie graphs.

E, F   Representative low and high magnification images of MTs nucleated for 10 min at 37 °C from 5 μM tubulin, in the presence of: 2 μM wild-type DCX (WT) (E) or 2 μM NDC-NDC (NN) (F).

G   Schematic of TIRF microscopy-based MT stabilization assay reported in H and I. Various concentrations of unlabeled DCX variants were added to 9 μM unlabeled tubulin mixed with 1 μM HiLyte Fluor™ 488-labeled tubulin in the presence of surface-immobilized HiLyte Fluor™ 488-labeled GMPCPP MT seeds; e-field, evanescent field.

H   Representative kymographs with arrowheads indicating MT catastrophes.

I   Quantification of MT catastrophe frequency in the presence of different DCX constructs observed with MT stabilization TIRF assay. Total free tubulin concentration was 10 μM in all experiments. The basal (tubulin alone) catastrophe frequency level at this tubulin concentration is indicated with a gray bar.

Data Information: In (D), data are presented as mean ± SD. In (I), data are presented as mean ± SD and smoothed curves connect the means. Number of MTs analyzed across 2–3 movies for each condition: 1 μM WT, $n = 26$; 0.5 μM WT, $n = 27$; 250 nM WT, $n = 26$; 50 nM WT, $n = 13$; 1 μM NN, $n = 27$; 0.5 μM NN, $n = 26$; 250 nM NN, $n = 25$; 50 nM NN, $n = 19$; N, $n = 18$; C1, $n = 24$; C2, $n = 22$; C3, $n = 18$; C4, $n = 20$; tubulin alone, $n = 29$. *$P = 0.03$; ****$P < 0.0001$ (one-way ANOVA).

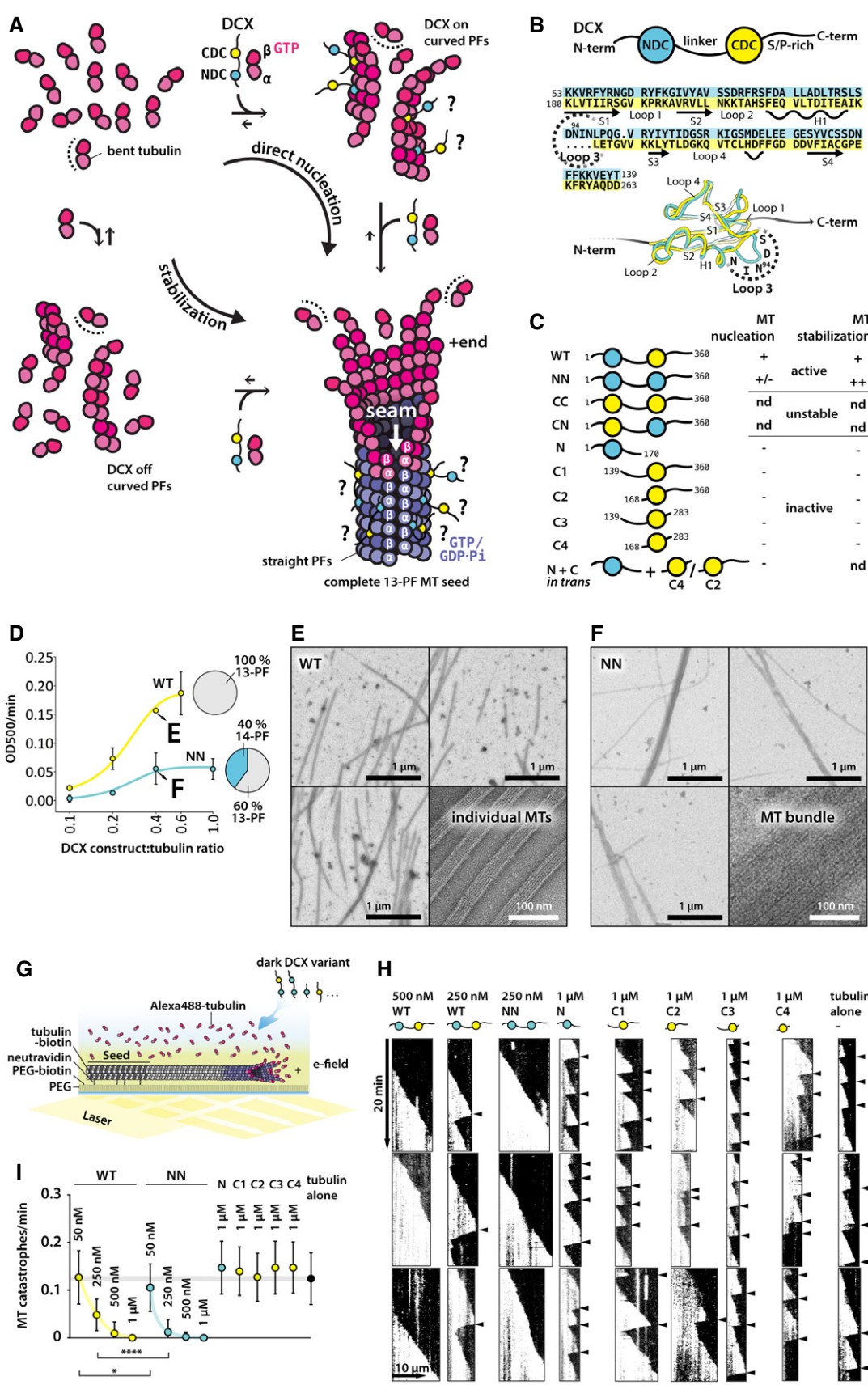

**Figure 1.**

two longest single-CDC constructs, C1 and C2 (Fig 1C, Appendix Fig S2C) also stimulated tubulin aggregation at high concentrations (3–5 μM; Appendix Fig S4).

Of the two tandem DCX constructs that could be analyzed further, WT appeared to be a more potent MT nucleator than NN (Fig 1D). It bundled MTs less than NN (compare Fig 1E and F), and nucleated solely physiological 13-PF MTs, as expected (Moores *et al*, 2004). In contrast, ~40% of MTs nucleated by NN had 14-PF architecture (Figs 1D and EV1A–C). This suggests that CDC in WT not only facilitates MT nucleation, but also participates in defining the resulting MT architecture (Fig 1A: direct nucleation).

### DC domain tandem is required for MT stabilization, with CDC not essential for this activity

Bulk MT polymerization cannot distinguish MT nucleation and stabilization activities of DCX. To uncouple MT nucleation from stabilization activity, we used MT seeds stabilized with GMPCPP, a slowly hydrolysable GTP analogue (Hyman *et al*, 1995). Stabilization of dynamic MTs grown from these seeds by different DCX constructs was monitored in a total internal reflection fluorescence (TIRF) microscopy assay using fluorescently labeled tubulin (Fig 1G). Single-DC domain constructs did not reduce MT catastrophe frequency, even at 1 μM concentration (Figs 1H–I and EV1D). Since C1 and C2 showed a tendency to aggregate tubulin (Appendix Fig S4), we also tested their MT stabilizing capabilities at lower concentrations (250 and 500 nM) and consistently observed no MT stabilization (Fig EV1D). Only WT and NN were capable of preventing MT depolymerization. NN completely abolished catastrophes at the concentration of 250 nM against 10 μM tubulin, whereas double that concentration (500 nM) of WT was required for the same effect (Fig 1H–I). At these sub-stoichiometric concentrations both DC domains in principle have unhindered access to their binding sites on the MT lattice. It is thus intriguing that NN has twice as many NDC domains as WT and requires half the concentration of WT to achieve the same MT stabilizing effects. This apparent gain of MT stabilization function upon substitution of NDC for CDC in NN suggests that NDC may have a higher affinity for MT lattice than CDC, and could also explain the potent MT bundling by NN (Fig 1F).

The ability of the DCX constructs to influence the frequency of MT rescues (resumption of growth after catastrophe) closely followed their ability to influence catastrophe frequency (Fig EV1E). MT growth rates were dose-dependently stimulated by NN and C1 (Fig EV1F). They were also significantly increased at higher concentrations (0.5–1 μM) of WT, N, and C2 (Fig EV1F). This contrasts with what has previously been reported (Moores *et al*, 2006; Bechstedt *et al*, 2014; see Discussion) and suggests that i) at high concentrations, even single-DC constructs are not completely inert regarding MT dynamic instability and ii) the properties of single-DC constructs are altered by the presence of adjacent regions of the DCX protein. The MT depolymerization rates were only significantly slower in the presence of double-DC constructs WT and NN, as quantified at lower concentrations (50–250 nM) where catastrophes still occurred (Fig EV1G).

### DCX binds growing MTs initially via CDC, which is subsequently replaced by NDC

Our experiments show that MT nucleation and stabilization activities of DCX can be differentiated and that each DC domain of DCX is involved in distinct facets of each process. To obtain high-resolution structural evidence of the mode of DCX involvement at different stages of MT assembly we turned to our previous cryo-EM study of MTs polymerized in the presence of WT for 30 s or 1 h. In these experiments, rapid (30 s) polymerization captured 13-PF MTs in a transient state that we initially interpreted as GDP.Pi-tubulin (Manka & Moores, 2018), but which could in fact be GTP-tubulin (Estévez-Gallego *et al*, 2020). One hour polymerization produced 13-PF GDP-MTs (Manka & Moores, 2018). In that previous study, we focused on changes in the MT lattice between these different nucleotide states (Manka & Moores, 2018), whereas here we focus on densities corresponding to DC domains bound to the MT lattice (Fig 2A–F, Appendix Fig S5 and Table 1). As before (Moores *et al*, 2004; Fourniol *et al*, 2010), no density could be assigned to regions of DCX other than the DC domain and immediately adjacent sequences, due to their flexibility.

We began assigning the DC domains in our reconstructions using the most pronounced difference between these domains, which is the length of Loop 3 (Fig 1B). Strikingly, the DC domain density in the rapidly polymerized WT-MT reconstruction most resembles

---

**Figure 2. NDC and CDC have distinct roles in stabilizing different stages of tubulin assembly.**

A   Representative low-pass filtered cryo-EM micrographs of MTs decorated with wild-type DCX (WT) or NDC-NDC construct (NN) after 30 s or 1 h incubation at 37 °C. Additional DCX proteins were added in excess after MT deposition on EM grids to maximize MT decoration. The final ratio of WT or NN to tubulin was 50:5 μM, giving dense protein background. WT co-polymerized exclusively 13-PF MTs, while NN both 13- and 14-PF MTs. Rapid (30 s) MT polymerization in excess of WT produces MTs in an intermediate GTP/GDP.Pi state, whereas long-lived WT-MTs and NN-MTs are all in the GDP state.

B   Isosurface +end and side views of cryo-EM reconstructions of MTs decorated with different DC domains: α-tubulin, light violet or light gray; β-tubulin, dark violet or dark gray; CDC, yellow; NDC, blue.

C   Close-up views of the DCX binding site focusing on loop 3 density, which differentiates DC domains. Refined atomic models in predominantly backbone representation are fitted to their respective density maps (wireframe). Selected residues are labeled with single-letter codes and selected side chains are shown as sticks. Nucleotide in tubulin subunit designated as β1 is shown as sticks in the top left corner of each panel. Colors as in (B), or oxygen atom, red; nitrogen atom, navy blue.

D   Cartoon illustration of the modes of WT or NN binding to MT lattice, based on cryo-EM results. WT decorates the GTP/GDP.Pi lattice predominantly via CDC ("C") or via both DC domains ("N-C") and the GDP lattice predominantly via NDC ("N") or via both DC domains ("N-C"). NN can decorate the GDP lattice via its N-terminal NDC ("N"), C-terminal NDC ("C"), or both ("N-C"). Modes "N" and "C" are compatible with MT cross-linking (bundling) by NN.

E–G  Assessment of the model-map fit. Upper panels show single-DC domains bound to four tubulin subunits: t, MT polymerization time; density maps, semi-transparent isosurface; models, backbone. Lower panels show close-up views of the MT-facing side of the DC domains (backbone + side chains shown as sticks). The main differentiating features between NDC and CDC are highlighted with dashed lines. NDC density in the NN reconstruction (G) results from averaging of two NDCs with varying N-terminal region, causing less defined density for that region (N-term scrambling, shown without side chains). Coloring as in (C).

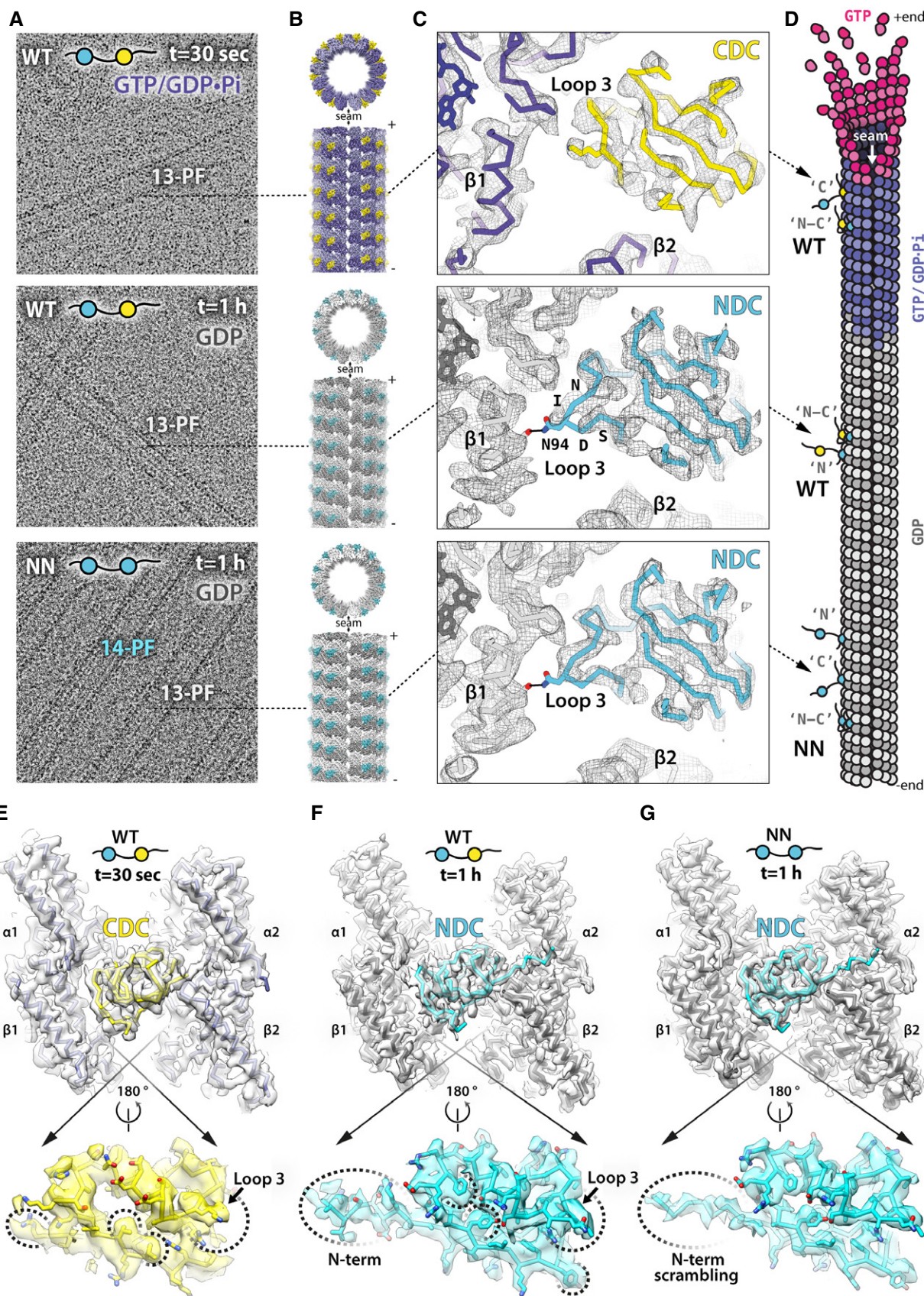

**Figure 2.**

**Table 1.  Cryo-EM data collection, atomic model refinement, and validation statistics**

| | WT(CDC)-GTP/GDP.Pi-MT 13-PF (EMD-4861) (PDB 6RF2) | WT(NDC)-GDP-MT 13-PF (EMD-4858) (PDB 6REV) | NN(NDC)-GDP-MT 13-PF (EMD-4862) (PDB 6RF8) | NN(NDC)-GDP-MT 14-PF (EMD-4863) (PDB 6RFD) |
|---|---|---|---|---|
| **Data collection and processing** | | | | |
| Magnification | 35,971 | 35,971 | 35,971 | 35,971 |
| Voltage (kV) | 300 | 300 | 300 | 300 |
| Electron dose (e-/Å$^2$) | 25 | 25 | 25 | 25 |
| Defocus range (μm) | −0.4 to −2.5 | −0.4 to −2.5 | −0.4 to −2.5 | −0.4 to −2.5 |
| Pixel size (Å) | 1.39 | 1.39 | 1.39 | 1.39 |
| Segment box size (pixels) | 652 × 652 | 652 × 652 | 652 × 652 | 432 × 432 |
| Symmetry imposed[a] | 12-fold | 12-fold | 12-fold | 13-fold |
| Initial no. of segments | 7,727 | 32,256 | 33,392 | 50,255 |
| Final no. of segments | 6,591 | 30,434 | 22,866 | 9,984 |
| Reconstruction software | Chuff | Chuff | Chuff | Relion 2.1 |
| Map resolution (Å) FSC$_{true}$ (0.143 threshold)[b] | 4.2 | 3.8 | 3.8 | 3.9 |
| **Refinement** | | | | |
| Initial models used (PDB) model:map fit[c] (cross-correlation) | 5IP4, 6EVX 0.80 | 1MJD, 6EVZ 0.84 | 1MJD, 6EVZ 0.85 | 1MJD, 6EVZ 0.86 |
| **RMS deviations** | | | | |
| Bond lengths (Å) | 0.004 | 0.005 | 0.005 | 0.005 |
| Bond angles (°) | 0.672 | 0.695 | 0.714 | 0.683 |
| **Validation** | | | | |
| MolProbity score | 1.27 | 1.27 | 1.25 | 1.25 |
| Clashscore | 1.88 | 1.48 | 1.7 | 1.8 |
| Poor rotamers (%) | 0 | 0 | 0 | 0 |
| **Ramachandran plot** | | | | |
| Favored (%) | 95.43 | 94.36 | 95.24 | 95.57 |
| Allowed (%) | 4.57 | 5.64 | 4.76 | 4.43 |
| Outliers (%) | 0 | 0 | 0 | 0 |

[a]DC domains do not bind at the MT seam, thus pseudo-helical symmetry was applied only across PFs between which DC domains bind: that is 12-PFs in 13-PF MTs and 13-PFs in 14-PF MTs.
[b]Gold standard resolution estimation according to (Chen *et al*, 2013).
[c]The refined models include a DC domain and four tubulin subunits that are in contact with that DC domain. The refinements are thus focused on the DCX-MT interfaces at different MT nucleotide states (for tubulin conformations at different MT nucleotide states, please see (Manka & Moores, 2018)). The model:map fit is calculated in an equivalent cryo-EM map fragment that the model has been refined against (see Materials and Methods). These models and maps are deposited in the PDB and EMDB databases with the identifiers indicated in the top row.

CDC, while the corresponding density in the 1 h MT reconstruction resembles NDC (Fig 2C, Appendix Fig S5). We thus refined atomic models of CDC and NDC together with the bound tubulin subunits in their respective density maps (Fig 2C and E and F and Table 1). Several differences in the CDC vs NDC sequences are also visible in the domain densities, in particular the relatively bulky side chains (Leu and Phe residues, Fig 2E and F, dotted lines), which further supports our domain assignment. An additional difference between the 30 s and 1 h reconstructions is that a short peptide corresponding to the sequence that precedes the DC domain is visible in the 1 h MT reconstruction, but not in the rapidly polymerized WT-MT reconstruction. We therefore assigned this density to the N-terminal

flanking region of NDC and modeled it accordingly (Fig 2E–F, Appendix Fig S5).

To verify this assignment, we carried out an analogous high-resolution 3D cryo-EM study of MTs polymerized for 1 h in the presence of NN. Comparison of the reconstruction of the NN-MTs—which can only have NDC bound to the MT—with the WT-MT reconstructions confirms that the 1 h WT-MT lattice is predominantly bound by NDC (Fig 2A–C and F and G, Appendix Fig S5 and Table 1). The NN-MT reconstruction shows somewhat more pronounced density for loop 3 than the WT-MT (Fig 2C), suggesting that there could be some mixing of the DC domains on the WT lattices (Fig 2D). On the other hand, the density for NDC's N-terminal flanking region is less

distinct in the NN-MT reconstruction than in the WT-MT (compare Fig 2F and G). Given that NN's two NDCs have different flanking regions, this implies that there is some mixing of these domains on the MT lattice (Fig 2D) and they appear to be averaged together (Fig 2G, N-term scrambling). The N-terminal domain cloning boundary of NDC is at residue 48 and the density starts to deteriorate upstream of this residue, where the sequence context of each NDC of NN starts to diverge. Taking all the comparisons together, we conclude that both the unique features of NDC—(i) its N-terminal flanking region, and (ii) its longer loop 3—are likely contributing to the observed predominance of bound NDC in 1 h WT-MTs.

## NDC binding to MT lattice is not selective for a particular MT architecture

Since NN-stimulated MT nucleation produces 14-PF almost as well as 13-PF MTs, we investigated whether there were any architecture-specific differences in NDC observed in these MTs. The manual collection of NN-MT cryo-EM data for high-resolution 3D reconstruction was focused on the physiological 13-PF architecture to enable direct comparisons with WT-MT reconstruction. Nevertheless, supervised 3D classification of all picked NN-MT segments revealed that 73% were 13-PF while 26% were 14-PF. The remaining 1% were particles of relatively poor quality. The resulting 14-PF NN-MT reconstruction had a resolution nearly equivalent (3.9 Å) to that of the 13-PF NN-MT reconstruction (3.8 Å; Fig 3A and B, Appendix Fig S5 and Table 1). Crucially, NDC density is equally pronounced in both the 13-PF and 14-PF NN-MT 3D maps (Fig 3A and B). Moreover, the level of decoration with NN of the 13/14-PF lattices is comparable to that of 13-PF lattices with WT, as evidenced by analysis of MT Fourier transforms (Fig 3C, Appendix Fig S6). NDC requires only minimal structural adaptation to fit to the slightly narrower inter-PF angle of the 14-PF lattice compared to the 13-PF lattice (Fig 3D). Overall, NDC appears to be relatively insensitive to the inter-PF angle, with identical (within the resolution) polar NDC-MT interactions formed in each MT architecture (Fig EV2).

## CDC and NDC have distinct binding footprints on the MT lattice

Using the calculated atomic models, we analyzed the CDC-MT and NDC-MT interfaces (defined as residues situated within 4 Å distance from tubulin; Fig 4). Both CDC and NDC interact with all four tubulin subunits at the lattice vertex, but with distinct tubulin residues. Overall, MT interactions by both NDC and CDC are mainly electrostatic, including several hydrogen bonds and/or salt bridges, but also involve hydrophobic contacts (Fig 4 and EV2). In 1 h WT-MTs, additional interactions are mediated by the N-terminal flanking region of NDC (residues 44–50; Figs 2F and 4). NDC refined in the WT-MT map and that refined in NN-MT map show almost identical contacts (Fig EV2). The longer Loop 3 of NDC is inserted in the inter-PF grove, where it more closely follows the shape of the MT wall than the shorter Loop 3 of CDC (Fig 2C). This endows NDC with an additional polar contact via N94 with β-tubulin's D209 and a hydrophobic contact via I95 with β-tubulin's A302 (Fig 4). Altogether, CDC forms 9 and NDC forms 11 ionic interactions with MT lattice (Fig EV2), which suggests higher affinity of NDC for MT lattice compared to CDC. Thus, our cryo-EM models suggest how NDC might be favored over CDC on the MT lattice.

## DC domain plasticity and its potential role in MT assembly

Is there a structural explanation for CDC being favored on the GTP/GDP.Pi-MTs and NDC on GDP-MTs? The interaction surface area of these domains with MTs is 1047 Å for CDC-GTP/GDP.Pi-MT and 1603 Å for NDC-GDP-MT (Fig 5A). However, the GTP/GDP.Pi lattice is locally similar to the GDP lattice at the DCX binding site, with the greatest deviation of tubulin backbone between the two states being at the α2 corner of the DC binding site, amounting to ~1 Å (Manka & Moores, 2018; Fig 5B). Therefore, there is no obvious mechanism by which the lattice itself could dictate preferential association of CDC with the transient GTP/GDP.Pi-MT state. We thus speculated that the lattice-based discrimination mechanism between NDC and CDC is a temporal one, such that CDC is involved in the early stages of MT formation and NDC in a later stabilization role.

Can the distinct properties of the DC domains shed further light on this idea? The conformation of the MT-bound CDC in our reconstruction is substantially different from that stabilized in its crystal structure (Burger *et al*, 2016; Fig 5C, yellow, root-mean-square deviation, RMSD = 4.3 Å). On the other hand, NDC shows structural rigidity regardless of the architecture of the MT to which it is bound (Fig 4), and this MT-bound conformation is highly similar to the published NMR and X-ray structures of NDC (Fig 5C, cyan, RMSD = 1.3 Å). We therefore propose that this apparent structural plasticity of CDC, and lack thereof in NDC, could be related to CDC's involvement in MT nucleation (Fig 1D). Specifically, this structural plasticity could enable CDC to both interact with curved tubulin assemblies early in MT formation (Figs 1A, EV3A and EV4) and be retained on the nascent MT lattice (GTP/GDP.Pi state; Figs 2B–E and 4). The subsequent favoring of NDC over CDC, due to the greater interaction surface of NDC, coincides with MT lattice maturation (GDP state). This can be visually appreciated by comparing the cross-sections of the CDC and NDC complexes with a particular tubulin maturation state (Figs 5C and EV3B).

## Pathogenic mutations in DC domains affect their fold or interaction with tubulin

Finally, we wanted to investigate how the known symptomatic DCX mutations in human correlate with MT binding interfaces established in this study. We used the OMIM[®] database (Online Mendelian Inheritance in Man: www.omim.org) (McKusick & Ruddle, 1987) to find all pathogenic and likely pathogenic DCX missense mutations, i.e., those that still produce the full-length protein, but with point mutations in amino acid sequence that compromise DCX function and cause a disease. We found 78 such mutations of which only 5 do not lie in the DC domains (Appendix Table S1). The 73 mutations that map to the DC domains involve 65 unique mutation sites, 33 in NDC, and 32 in CDC (Fig 6, red circles). Of the NDC mutation sites, 21 involve residues that contribute to the domain fold (Fig 6A, white), and 12 are at the MT binding interface (Fig 6A, green). Of all CDC mutation sites, 21 involve core residues that determine the domain fold (Fig 6B, white), seven are at the MT binding site (Fig 6B, green with straight numbers) and four are functionally unassigned (Fig 6B, green with *italicized* and shifted numbering). These unassigned residues may be involved in interaction with curved tubulin assemblies or in determining domain plasticity. The mapped 21 core residues in CDC are different than the mapped 21 core

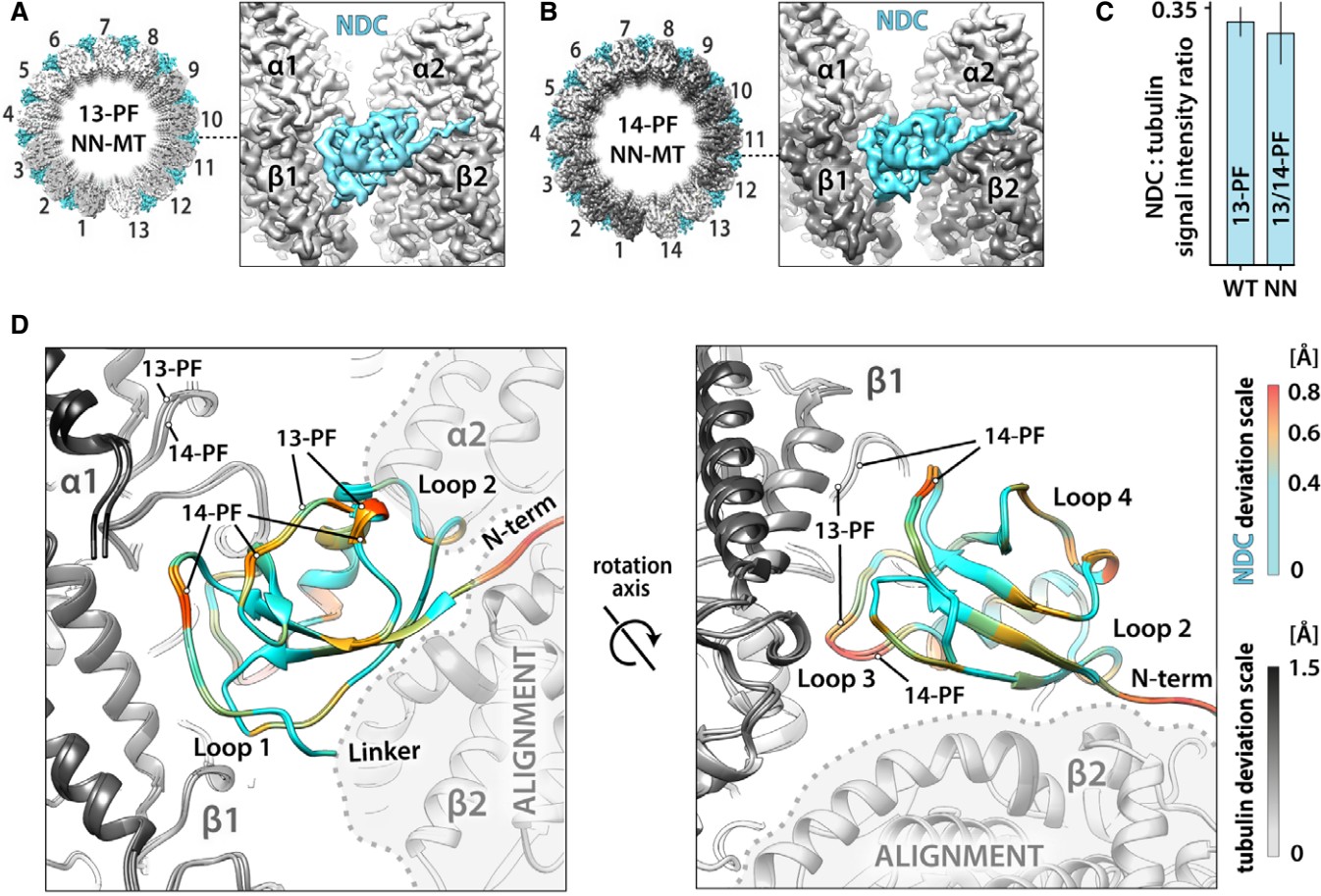

**Figure 3. NDC proficiently recognizes and stabilizes both 13-PF and 14-PF MTs.**

A  Isosurface top view of the whole 13-PF MT cylinder and a side view close-up on NN binding site at the vertex of four tubulin dimers, where NDC of NN decorates the MT lattice. α-tubulin, white; β-tubulin, gray; NDC, blue. The valley between PFs 1 and 13 represents MT seam, where NN does not bind.

B  Isosurface views as in (A) of the 14-PF MT decorated with NN. α-tubulin, light gray; β-tubulin, dark gray; NDC, blue. The valley between PFs 1 and 14 represents MT seam, where NN does not bind.

C  Relative extent of MT lattice decoration based on average intensities of layer lines representing NDC (8 nm periodicity) and tubulin (4 nm periodicity) in averaged Fourier transforms of WT-MT and NN-MT segments, as described in Appendix Fig S6 and in Materials and Methods.

D  Comparison of the DCX binding clefts in the 13-PF and 14-PF MTs and mapping of the associated small (<0.8 Å) conformational adjustments in NDC. NN-MT lattice fragments were aligned on the indicated α2 and β2 tubulin subunits (ALIGNMENT). Deviations in tubulin are depicted with gray scale and in NDC with color scale. Two perspectives are shown to cover all major MT-binding loops of NDC.

Data Information: In (C) data are presented as mean ± SD of three approximately equal, non-overlapping subsets of data into which WT-MT segments (*n* = 30,434 total) and NN-MT segments (*n* = 32,850 total) were divided.

residues in NDC and they exhibit different distribution within the domain (Fig 6). Thus, while it is well established that DCX mutations can cause disease by either disrupting DCX binding to MT or by disrupting the DC domain fold (Kim *et al*, 2003), our data highlight that disruption of domain plasticity may also be a disease mechanism arising from mutations specifically in CDC.

# Discussion

Doublecortin is critical for neuronal motility during mammalian brain development. It localizes to the distal processes of the migrating immature neurons (Tint *et al*, 2009), where the absence of the centrosomal γ-tubulin ring complexes (γ-TuRCs) that normally

template the 13-PF architecture, may require that DCX contributes to *de novo* 13-PF MT nucleation to help direct cell migration (Stiess *et al*, 2010). In this work, we dissect the roles of the two pseudo-repeats (DC domains) in DCX, thereby providing mechanistic insight into how this neuronal migration protein nucleates and stabilizes MTs (Fig 7). Doublecortin needs both DC domains for proper function, but our data suggest that a 2-step mechanism is involved, where each step is served by distinct DC domains. Step 1 is mediated by CDC and pertains primarily to MT nucleation, while step 2 is mediated by NDC as well as CDC and pertains to longer-term MT stabilization. Crucially, our experiments show that each DC domain can fulfill its entire dedicated role only as a part of the DCX tandem, agreeing with earlier studies (Horesh *et al*, 1999; Taylor *et al*, 2000). We observed stimulation by certain single-DC domain

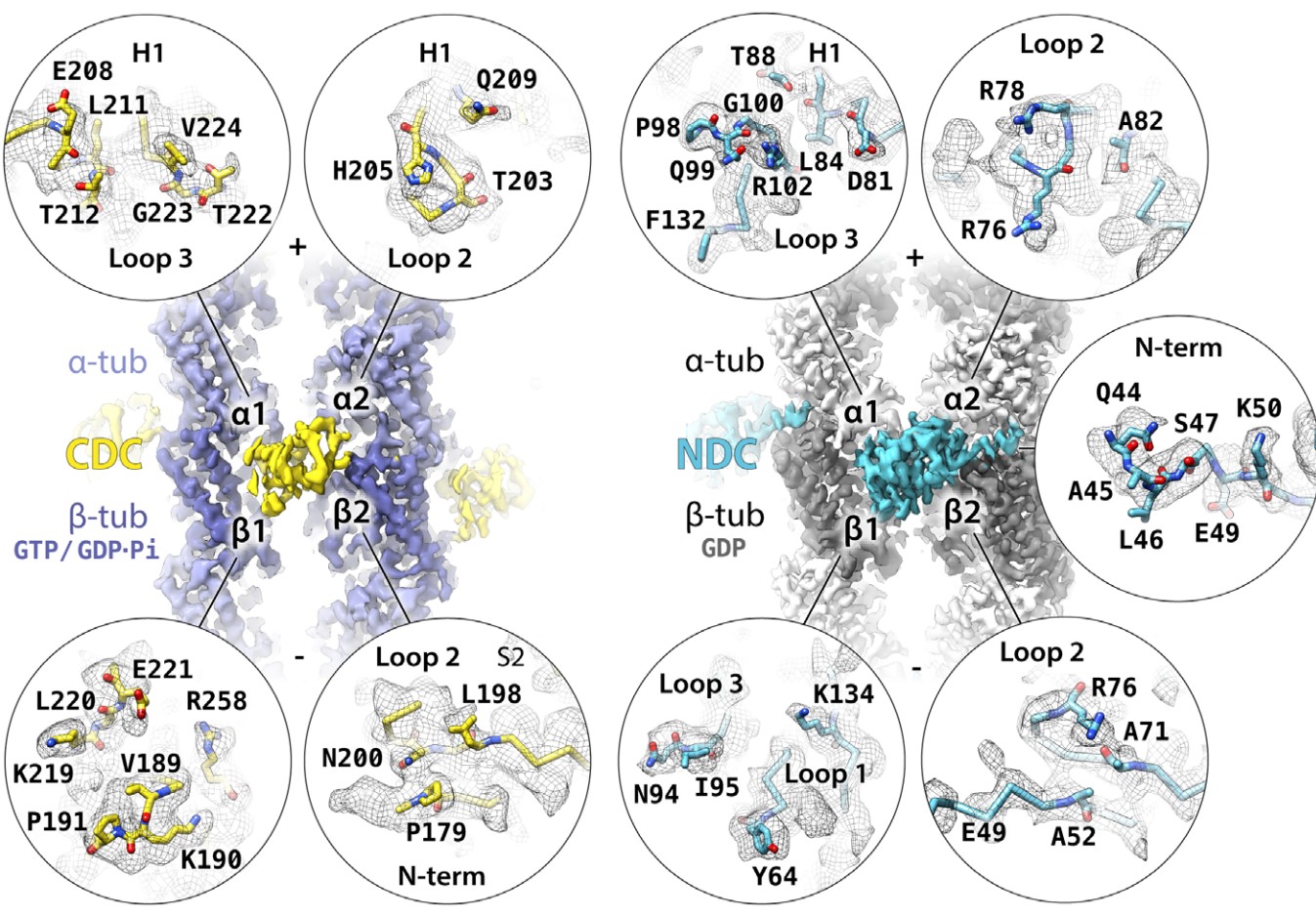

**Figure 4. MT-binding residues of DCX.**

Isosurface overviews of single-DC domains bound to MT lattices together with detailed presentations of all DC domain residues interfacing with MT (4 é distance), shown in circles representing individual tubulin subunits or binding site of the N-terminal flanking sequence of WT's NDC. Only DC residues with at least one atom found within the 4 é contact zone are fully represented together with their respective densities (wireframe). The density is not masked around the selected residues and occasionally includes adjacent residues that are not displayed in the model. The contact residues are viewed from MT side, i.e., flipped horizontally with respect to the isosurface overviews. α-tubulin, light violet or light gray; β-tubulin, dark violet or dark gray; CDC, yellow; NDC, blue; oxygen atom, red; nitrogen atom, navy blue.

constructs on MT growth rates; in addition, and contrary to earlier studies (Moores *et al*, 2006; Bechstedt *et al*, 2014), the wild-type DCX (WT), as well as the NDC-NDC tandem (NN), increased MT polymerization rates at high concentrations, albeit modestly compared to bona fide polymerases such as XMAP215 (Brouhard *et al*, 2008). The discrepancies between our current work and previous studies likely stem from combinations of the differences between DCX isoforms (Fig EV5A) and protein expression systems used in different studies (see further discussions).

We propose that CDC in DCX can directly nucleate MTs, because we found that its substitution for another NDC: (i) slows down MT nucleation, and (ii) results in a heterogeneous MT population (13- and 14-PF MTs). It has previously been proposed that DCX can recognize longitudinal curvature of MTs via CDC (Bechstedt *et al*, 2014), and this could also underlie the interaction with early, curved tubulin assemblies (Figs 1A: direct nucleation, and EV4), leading to generation of complete MT seeds for subsequent elongation (MT growth). Thus, CDC might be specifically involved in stabilizing such early intermediates of MT nucleation, due to its conformational

flexibility, e.g., binding bent tubulin assemblies via one conformation and the early (GTP/GDP.Pi) MT lattice via another. Retention of CDC at the vertex of four tubulin dimers could help stabilize lateral and longitudinal contacts between these dimers (Fig 7B). It might also limit longitudinal flexibility (bending) of the assembly (Figs 7B and EV3A), facilitating lattice closure. This tantalizing hypothesis is supported by the fact that no MTs were observed when tubulin polymerization in the presence of NN for 30 s was attempted, while a relatively large number of curved pre-MT tubulin oligomers were still present after 1 h-long incubation with NN (Fig EV4). Thus, we suggest that MT nucleation by NN is inefficient and architecturally inaccurate compared to WT specifically because of the absence of CDC's interaction with early tubulin nuclei (Fig 1D). The high conformational stability of NDC compared to CDC, on the other hand, offers an explanation for its apparent role as an MT stabilizer.

Our results suggest that CDC may remain associated with tubulin oligomers while they undergo the dynamic transition from bent to straight conformation, to become a part of MT lattice. The molten globule properties of CDC were previously characterized using NMR

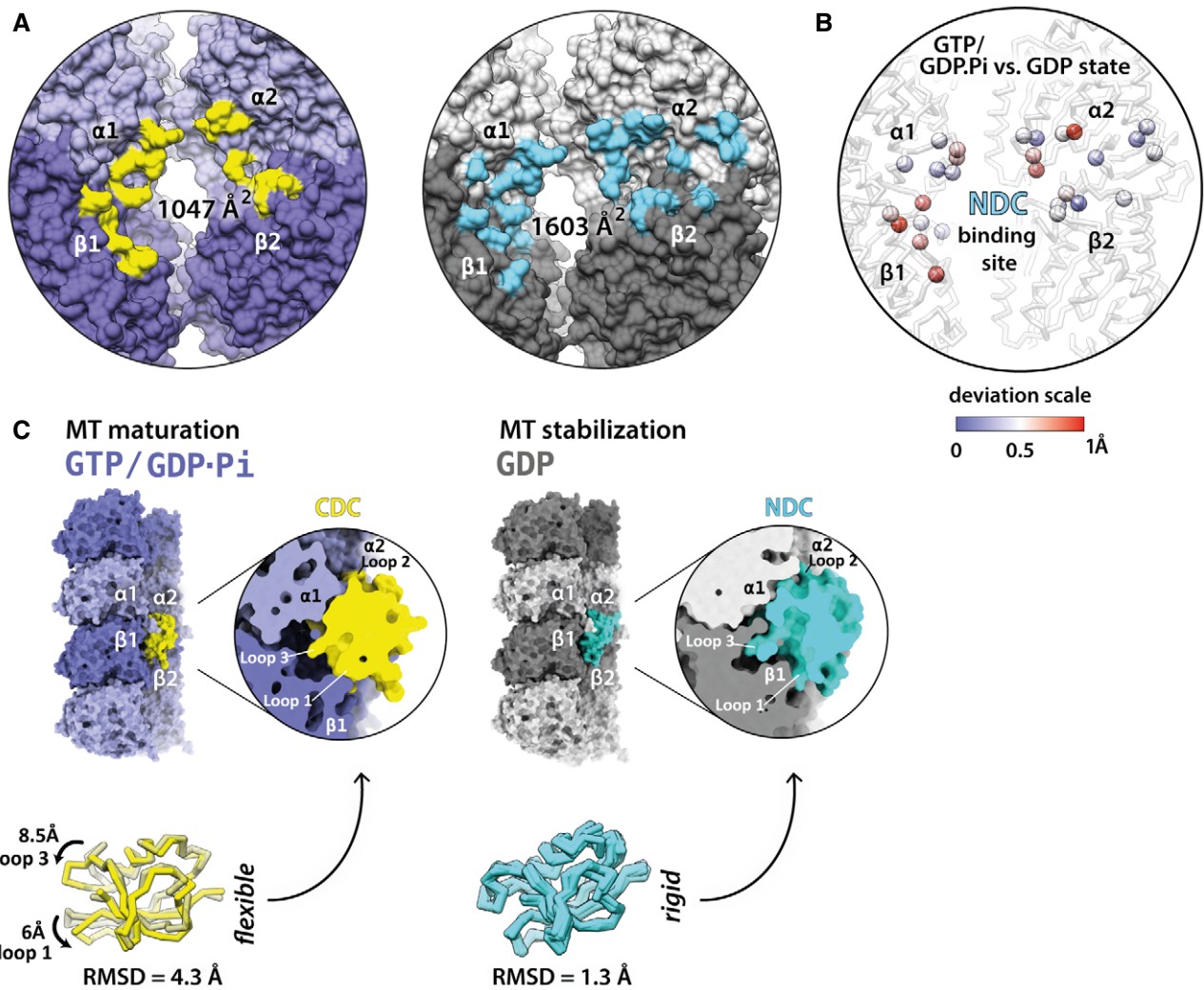

**Figure 5. MT binding by DC domains according to their footprint and conformational plasticity.**

A   DC domain binding footprints (residues within 4 Å distance) on MT lattice at GTP/GDP.Pi and GDP states. Interface areas were calculated using PDBePISA (http://www.ebi.ac.uk/pdbe/pisa/).

B   Differences between NDC binding sites in GTP/GDP.Pi- and GDP-MT lattices. Deviations in backbone (Cα) atom positions are illustrated with color-coded spheres.

C   Top, binding of CDC and NDC at the vertex of four bent tubulin dimers of different MT nucleotide states with close-up cutaway views. Bottom, superposition of CDC X-ray structure 5IP4 (darkened and transparent) with the GTP/GDP.Pi-MT-binding conformation (this study; bright yellow) and superposition of five NMR and X-ray NDC structures from Protein Data Bank (PDBs: 5IO9, 1MJD, 2BQQ, 5IKC, 5IN7) with the GDP-MT-binding conformation (this study).

(Kim *et al*, 2003), while CDC has only recently been crystallized in complex with a stabilizing antibody (Burger *et al*, 2016) or as a domain-swapped dimer (Rufer *et al*, 2018), also reflecting the domain's flexible nature. Consistently, our CDC-CDC tandem (CC construct) proved to be unstable at 37 °C, and the other construct having CDC near the N-terminus (CN) showed malfunctions that can likewise be attributed to potential folding defects, including low level of expression (e.g., issues with solubility; Appendix Fig S3) and a tendency to aggregate tubulin (Appendix Fig S4). These results also hint that the precise context of CDC within the DCX protein may contribute to its properties.

CDC's role as a quality control agent, tuning MT assembly to produce exclusively 13-PF MTs, may be because it can limit lateral as well as longitudinal conformational variability in tubulin nuclei. The C-terminal S/P-rich domain was previously found to

also have a regulatory role in determining DCX-mediated MT architecture (Moores *et al*, 2004), an effect that may be direct or mediated via CDC. Here, we observed MT growth stimulation by CDC-S/P-rich constructs (C1 and C2), but not by an isolated CDC, devoid of the S/P-rich domain (C3 and C4). This further emphasizes the apparent regulation of DC domain function by DCX's disordered C-terminal region, while the instability and low activity of the CDC-NDC chimera point to the importance of the sequence context in which each domain is located. The mechanisms of these effects, especially in the context of DCX isoforms and the presence or absence of post-translational modifications (PTMs), are not understood and will be the focus of future work. Overall, we conclude that the remarkable conformational plasticity of CDC reflects the conformationally dynamic nature of its oligomeric tubulin substrate during MT nucleation.

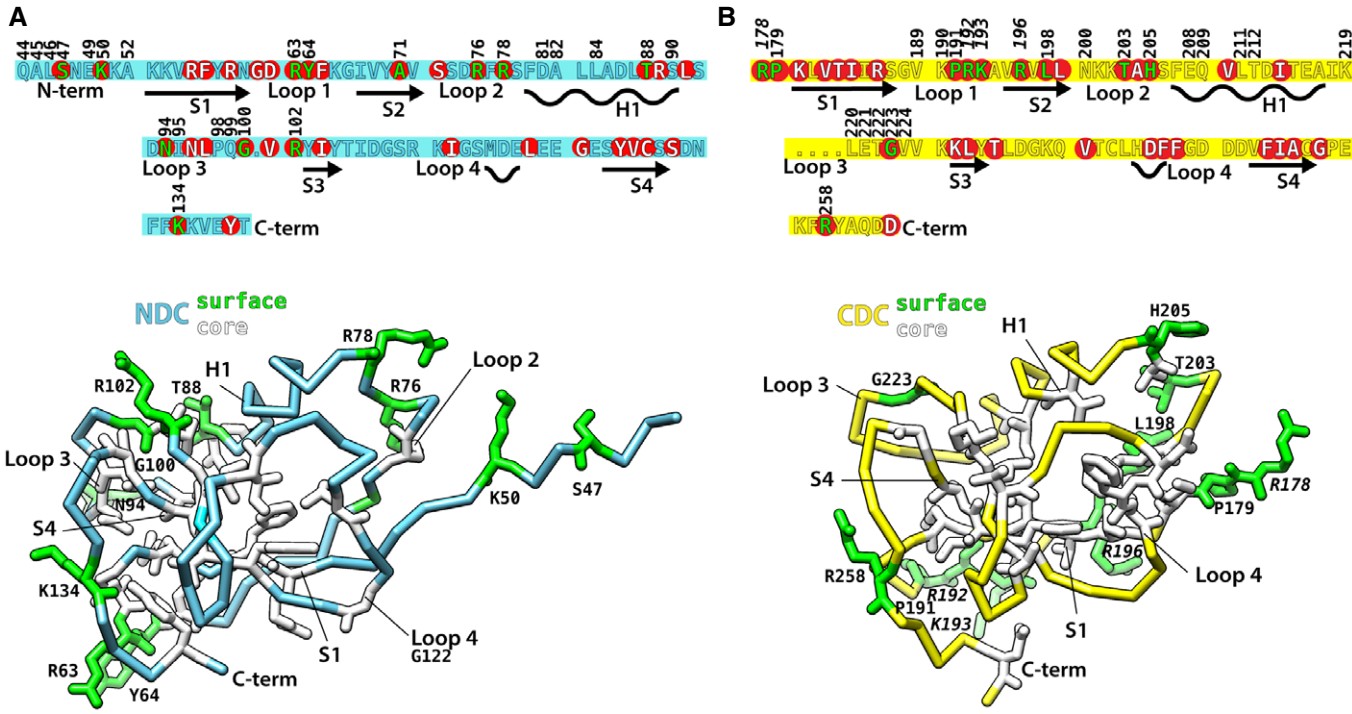

**Figure 6. Disease-causing mutations in DC domains map to their core or to DCX-tubulin/MT interface.**

A  The human NDC sequence and the cryo-EM structure of the MT-bound conformation, including the short N-terminal flanking region. Pathogenic mutation sites (Appendix Table S1) are indicated in the sequence with red circles. Side chains of residues representing the mutation sites are depicted in green if they contribute to protein surface (mediate or likely mediate MT binding) or white if they lie in the domain core. MT-binding residues are numbered, and secondary structures are labeled. H, α-helix; S, β-strand.

B  The human CDC sequence and the cryo-EM structure of the MT-bound conformation. Pathogenic mutation sites (Appendix Table S1) are indicated in the sequence and the 3D structure as in (A). MT-binding residues are numbered using normal font immediately above the sequence and those unassigned are indicated with numbering shifted away from the sequence and *italicized*.

We identified NDC as the primary stabilizer of the MT lattice after CDC-mediated nucleation. It appears to join CDC along the MT shaft shortly after nucleation, but at large DCX excess over available MT binding sites, NDC appears to replace CDC, suggesting that it has a higher MT lattice binding affinity than CDC. This is consistent with the more extended MT-binding interface of NDC compared to CDC (summarized in Fig 7C) due to both the co-binding of the short pre-NDC region and a deeper penetration of the lattice with the longer loop 3. Physical linkage with CDC ensures proximity of NDC after MT nucleation, facilitating the DC domain enrichment on MT lattice during MT maturation (Fig 7A). In theory, DCX binding to MT via both DC domains results in a much stronger interaction between each DCX molecule and MT lattice (Fig 7C). This would predict that in a certain DCX concentration range, the lower the stoichiometry of DCX on the MT lattice, the tighter its affinity. Conversely, the cooperative DCX binding to MT, described by the Brouhard lab (Bechstedt & Brouhard, 2012), means that DCX dissociates from MT lattice more slowly as the DCX concentration increases. This may mean that the cooperative DCX binding mechanism may not be mediated by the DCX molecule itself, but rather via MT lattice effects exerted by DCX, such as the lattice regularization that may optimize DC domain binding sites.

Unlike CDC, NDC has been crystallized multiple times (Kim *et al*, 2003; Cierpicki *et al*, 2006; Burger *et al*, 2016), and superposition of

these X-ray structures with our cryo-EM structure suggests high conformational stability of NDC, which suits its proposed role as an MT anti-catastrophe agent. Such rigidity combined with the precise fit to MT lattice vertex prevents PFs from curling outward and breaking apart. Each such vertex is built from four tubulin dimers, and—under saturating conditions—each tubulin dimer in the lattice can be held by four NDC domains. Despite being relatively conformationally rigid, we found NDC still capable of adapting to slightly varying inter-PF angle, as evidenced with equal decoration of 13- and 14-PF MTs with NDC-NDC tandem (NN construct). This property of NDC—enabled primarily through loop flexibility—has not previously been reported, although the first low-resolution cryo-EM reconstruction of MT decorated with C-terminally truncated DCX (tDCX, lacking the S/P-rich region) clearly showed a DC domain binding to the 14-PF lattice (Moores *et al*, 2004). The potential physiological significance of this adaptation is not known.

Our sequential mechanism of MT nucleation and stabilization by DCX is overall consistent with earlier work (Horesh *et al*, 1999; Sapir *et al*, 2000; Taylor *et al*, 2000; Moores *et al*, 2004, 2006; Fourniol *et al*, 2010; Bechstedt & Brouhard, 2012; Liu *et al*, 2012; Bahi-Buisson *et al*, 2013; Bechstedt *et al*, 2014; Ettinger *et al*, 2016). Previous low- and intermediate-resolution cryo-EM studies proposed NDC as the predominant MT-binding domain (Moores *et al*, 2004; Fourniol *et al*, 2010; Liu *et al*, 2012). Using 366 amino acid (aa)

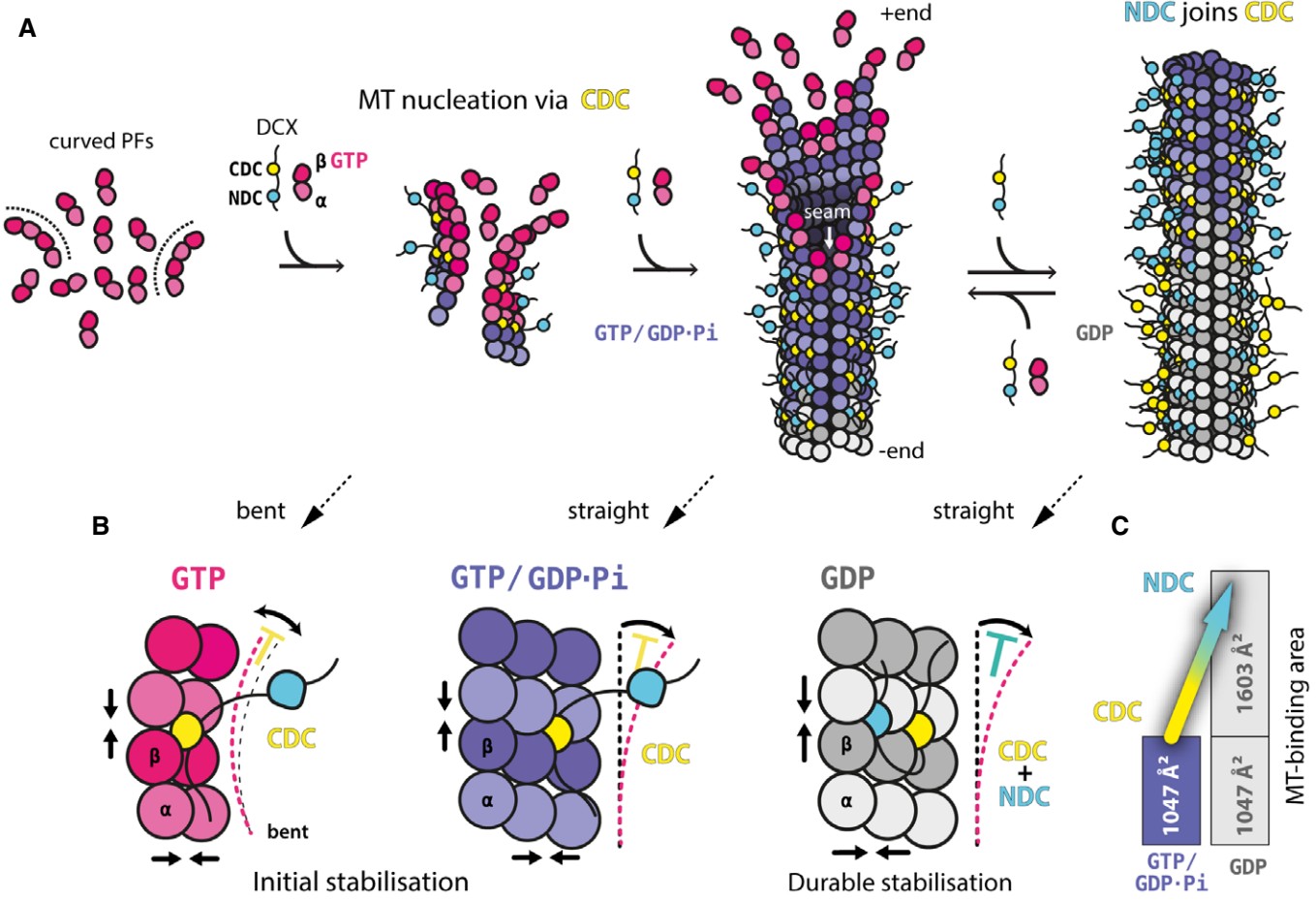

**Figure 7. Mechanism of MT nucleation and stabilization by DCX.**

A DCX nucleates MTs directly by stabilizing early tubulin oligomers (curved PFs). CDC mediates this activity as well as the initial stabilization of the nascent GTP/GDP.Pi-MT lattice. Then NDC joins and/or replaces CDC along the mature GDP-MT lattice.

B Summary of the sequential involvement of the two DC domains at different stages of MT assembly, characterized by different (bent or straight) tubulin conformations and different nucleotide states. Straight arrows indicate stabilization of tubulin-tubulin contacts. The color-coded T shapes suggest DC domain-dependent inhibition of structural change: CDC binding between bent tubulin dimers may restrict longitudinal curvature of the assembly in addition to stabilizing the lateral and longitudinal contacts between tubulin dimers, and the subsequent CDC and NDC binding to MT lattice appears to lock tubulin in the lattice-specific straight conformation, preventing MT depolymerization associated with straight-to-bent tubulin transition.

C Summary of the DCX-MT binding area at different stages of MT maturation.

isoform of human DCX (a C-terminal splice variant (Fig EV5A)) expressed in eukaryotic cells, the Moores lab visualized DCX density bound to MTs that resembled NDC (including the pre-NDC flanking region) in the presence of a kinesin motor domain (Fourniol *et al*, 2010; Liu *et al*, 2012). However, decoration with DCX alone showed no density for the pre-NDC flanking region but did reveal docking of the post-NDC linker region along the NDC core (Liu *et al*, 2012), likely via W146 as shown earlier in solution (Cierpicki *et al*, 2006). In our current study using bacterially expressed 360 aa human isoform 2 (Fig EV5A), we did not observe such linker docking, which may be due to the isoform differences and/or the lack of PTMs that may play additional regulatory roles.

A co-pelleting study using MTs pre-stabilized by Taxol reported that only a CDC-specific, and not an NDC-specific, antibody prevented DCX from binding MTs, suggesting that it is primarily CDC that mediates the DCX-MT interaction (Burger *et al*, 2016).

However, we previously found that Taxol MT stabilization reduces subsequent DCX binding (Manka & Moores, 2018), while others have found that addition of Taxol strips DCX (expressed in physiological levels) from MTs in cells, except from curved MT regions (Ettinger *et al*, 2016). Our new data suggest that Taxol MTs may support CDC—but not NDC—binding to MTs through its recognition of bent or flexible MT lattice regions, an interaction that is inhibited by anti-CDC antibodies. This would also be consistent with recent observations describing the role of CDC in the interaction of the DCX relative DCX-like kinase (DCLK) with Taxol MTs (Rogers *et al*, 2020). The apparent inhibition of NDC binding to straight MT lattice by Taxol is enigmatic, but—as we previously suggested (Manka & Moores, 2018)—may be related to the drug's mechanism of MT stabilization, producing a looser MT lattice and more flexible MTs (Kikumoto *et al*, 2006; Kellogg *et al*, 2017).

Our analysis of the disease-linked missense mutations in the DC domains reveals that these mutations affect either the domain fold or the hereby determined DCX-MT interaction for NDC and for CDC only in part. Some of the mutation sites on the surface of CDC remain functionally unassigned. These sites might mediate recognition of bent tubulin assemblies or determine structural plasticity of CDC. Interestingly, the 65 sites in the DC domains identified as essential for DCX function appear to be completely conserved across the vertebrate classes, in which forebrain is the dominant part of the brain: from mammals, through birds and reptiles, to amphibians (Fig EV5B–D). We found divergence in six of these residues in fish, the lowest vertebrate, where the midbrain is the dominant part of the brain: three in NDC and three in CDC (Fig EV5B–D). This suggests that the observed adjustments in the sequence of the DC domains may have a role in enabling forebrain development in higher vertebrates, potentially contributing to wider molecular synergies in this part of the brain and elaboration of the mammalian neocortex (Briscoe & Ragsdale, 2018).

The characterization of distinct roles for each DC domain within DCX reflects the conformational diversity of tubulin as it transitions between unpolymerized and polymerized states (Brouhard & Rice, 2018), and is influenced by MT aging, damage, and repair (Schaedel *et al*, 2015). In this context, different domains within MAPs effectively act as stage-specific tubulin chaperones specializing in interactions with different tubulin conformations. This is evident even in evolutionarily distant organisms, such as, for example, Toxoplasma gondii, where a DC domain containing protein TgDCX is essential to induce curvature in tubulin polymers (Leung *et al*, 2020). The presence of DC domain pseudo-repeats with different functional specializations supports multiple activities in DCX, including acceleration of MT nucleation, control of MT architecture, MT stabilization, and interaction with other components of MT cytoskeleton (Liu *et al*, 2012), all within a single molecule. It is also possible that DCX facilitates MT repair by chaperoning GTP-tubulin oligomers destined for incorporation into the site of MT damage via CDC and by subsequently stabilizing the repair site via NDC.

The presence of pseudo-repeats is common in MAPs, for example, the TOG domains in XMAP215/CLASPs (Slep, 2018; Cook *et al*, 2019), STOP or MAP2/tau family of MAPs containing 3–4 pseudo-repeats (Feng & Walsh, 2001). This reflects the importance for many of these proteins of (transiently) sensing or stabilizing a conformational range of tubulin ensembles undergoing dynamic transitions. Such modular MAPs can also favor or disfavor interactions with other MAPs, motors (kinesin and dynein), or MT-modifying enzymes (Monroy *et al*, 2020). They also regulate MT bundling or cross-linking with other cytoskeletal assemblies such as actin filaments. It will be interesting to evaluate the differential roles of pseudo-repeat containing binding proteins in other cytoskeleton filament systems in the light of our findings.

# Materials and Methods

## Generation and cloning of DCX variants

Doublecortin constructs were based on human isoform 2 sequence (Uniprot identifier: 043602-2)—here called wild-type (WT)—and designed through inspection of the available crystal structures. The truncated variants were generated using standard PCR methods from the gene sequence kindly gifted by Anne Houdusse (Institute Curie, Paris). Constructs were subcloned into pNic28-Bsa4 vector (Structural Genomics Consortium, Oxford, UK), with an N-terminal tobacco etch virus (TEV) protease-cleavable His-tag. The DC domain duplication and swapped chimeras were obtained with a multistep PCR amplifications and restriction enzyme reactions (New England Biolabs) as depicted in Appendix Fig S1.

## Purification of DCX variants

All DCX constructs were expressed in One Shot™ BL21 Star™ (DE3) Chemically Competent *E. coli* cells (Thermo Fisher) at 18 °C. The cells were spun and resuspended in ice-cold lysis buffer (50 mM $Na_2HPO_4$ pH 7.2, 300 mM NaCl, 10 mM imidazole, 10% glycerol, 2 mM DTT) supplemented with protease inhibitor cocktail (cOmplete Cocktail Tablet, Roche/Sigma Aldrich). The cells were lysed by sonication. The lysates were clarified by centrifugation and passed through a nickel HisTrap HP column (GE Healthcare). The His-tagged proteins were then eluted with 10–250 mM imidazole gradient. To cleave off the tags we used a His-tagged TEV protease expressed in-house and both the cleaved His-tag and the His-tagged protease were removed from protein solutions by passage over loose Ni-NTA His-bind resin (Merck). The tag-free DCX variants were then captured on a HiTrap SP HP ion exchange column (GE Healthcare) equilibrated in BRB80 buffer (80 mM PIPES [piperazine-N,N′-bis(2-ethanesulfonic acid)] pH 6.8, 1 mM EGTA [ethylene glycol-bis(β-aminoethyl ether)-N, N,N′,N'-tetraacetic acid], 1 mM $MgCl_2$, 1 mM DTT [dithiotreitol]) and eluted with NaCl gradient (15–300 mM). Final purification and desalting were done by gel filtration through Superdex 200 size exclusion column (GE Healthcare) equilibrated in BRB80 buffer.

## Thermal shift protein stability assay

The stability of DCX constructs was assessed with temperature denaturation monitored by SYPRO Orange dye that becomes highly fluorescent on contact with exposed hydrophobic regions (ThermoFluor). The 5000X SYPRO Orange dye solution in DMSO (Life Technologies) was diluted 250 times in BRB80 buffer containing 5 μM DCX construct. The change in fluorescence was measured in a 96-well plate by MyIQ RT–PCR instrument (BioRad). The melting temperature ($T_m$) is at the inflection point of the upward curve and is conveniently identified with a peak of a derivative function (Appendix Fig S3).

## Turbidimetric MT nucleation assay

Microtubule nucleation can be monitored by the extent of light scattering or turbidity. Tubulin (lyophilized bovine brain tubulin, >99% pure, Cytoskeleton, reconstituted in BRB80 buffer supplemented with 1 mM GTP at 5 μM concentration was mixed with 1–5 μM DCX construct in BRB80 buffer in an Eppendorf UVette (220–1,600 nm). The increasing sample turbidity was measured at 500 nm wavelength over 30 min at 37 °C with Cary 3 UV/VIS spectrophotometer equipped with a temperature control unit (Varian).

## Negative stain EM

Samples removed directly from turbidimetric assays were deposited on EM grids with a continuous carbon film (Agar). The grids were briefly blotted and washed with BRB80 buffer before staining with 1% uranyl acetate solution in water (negative stain solution). After ~1 s exposure to uranyl acetate, the grids were blotted and air-dried. The negatively stained grids were imaged on a 120 kV Tecnai T12 microscope (FEI) with a US4000 4K × 4K CCD camera (Gatan).

## TIRF microscopy assays

All TIRF microscopy assays were performed in ~10 µl flow chambers built through attachment of biotin-PEG coverslips (Stratech) to glass slides with a double-sided tape. The chambers were blocked by 5 mg/ml casein solution in 0.75% Pluronic F-127, washed with 0.4 mg/ml casein solution in BRB80 buffer (Wash solution), coated with 0.5 mg/ml neutravidin solution in BRB80 buffer, and washed again with Wash solution before application of GMPCPP-MT seeds. The fluorescent and surface-attaching (neutravidin-binding) GMPCPP-MT seeds were polymerized from a mixture of 10 µM unlabeled tubulin, 10 µM HiLyte Fluor™ 488-labeled tubulin, 10 µM biotin-tubulin (all from Cytoskeleton, >99% pure), and 1 mM GMPCPP (Jena Biosciences) in BRB80 buffer, incubated at 37 °C for 30 min.

For MT stabilization assay, ~0.1 µM GMPCPP-MT seed solutions in BRB80 (according to tubulin dimer concentration) were applied to flow chambers. After washing with Wash solution the chambers were ready to receive an assay solution comprising: 9 µM unlabeled tubulin, 1 µM HiLyte Fluor™ 488-labeled tubulin, 0.4 mg/ml casein, 1 mM GTP, 5 mM DTT, 20 mM glucose, 0.1% methyl cellulose, 300 µg/ml glucose oxidase and 60 µg/ml catalase, with varying concentrations of unlabeled DCX variant in BRB80 buffer.

Microtubule dynamics were observed using an Eclipse Ti-E inverted microscope equipped with a CFI Apo TIRF 1.49 N.A. oil objective, Perfect Focus System, H-TIRF module, LU-N4 laser unit (Nikon), and a quad-band filter set (Chroma). Exposures were recorded over 100 ms with 2-s intervals or continuously on an iXon DU888 Ultra EMCCD camera (Andor), using NIS-Elements Software (Nikon). Kymographs of dynamic and DCX-stabilized MTs were generated in FiJi (https://fiji.sc/) (Schindelin *et al*, 2012).

## TIRF microscopy data analysis

Microtubule dynamic instability parameters (Figs 1I and EV1D–G) were quantified from kymographs using a FiJi (https://fiji.sc/) macro (Roberts *et al*, 2014), available at: https://doi.org/10.7554/elife.02641.017. The obtained values were plotted and statistically analyzed with Prism 8 (graphpad.com). Details of all statistical methods and tests used are provided in the relevant figure legends. Sample size was determined based on the data scatter and the magnitude of the difference between the compared effects. The achieved statistical power is reflected in the stringent significance level (up to $P < 0.0001$), thus the number of subjects is sufficient, but usually much greater than sufficient, wherever statistical significance is stated. We did not exclude any data in these experiments. We treat all apparent outliers within each movie and between repeated movies as variations due to chance (biological variability)

and not due to random technical errors, therefore, we decided that their removal would be unjustified.

## Cryo-EM sample preparation

Tubulin stock was prepared by reconstituting lyophilized bovine brain tubulin (Cytoskeleton, >99% pure as above) to 100 µM concentration in BRB80 buffer supplemented with 1 mM GTP.

For GTP/GDP.Pi-MTs 10 µM GTP-tubulin was mixed with 50 µM WT in cold BRB80 buffer containing 1 mM GTP. The mixture was immediately applied to a glow-discharged Lacey grid (Agar) and transferred to a Vitrobot (FEI/Thermo Fisher Scientific) for rapid on-grid MT polymerization in a humid and warm (30 °C) Vitrobot chamber. After 30 s incubation, the grid was blotted and plunge-frozen in liquid ethane.

For GDP-MTs 5 µM GTP-tubulin was co-polymerized for 30 min at 37 °C with 3 µM WT or NN in BRB80 buffer containing 1 mM GTP. These MTs were applied to glow-discharged Lacey grids and incubated for 20 s at room temperature. Then the grids were briefly blotted and 50 µM WT or NN solution in BRB80 buffer was added to maximize MT decoration. The grids were then transferred to the Vitrobot and incubated there for 1 min at 30 °C, before being blotted and vitrified as before.

## Cryo-EM data collection

Cryo-micrographs were acquired on a 300 kV Polara microscope (FEI) with a K2 Summit camera (Gatan) operated in counting mode after a Quantum energy filter (Gatan) with a 20 eV slit. The magnified pixel size was 1.39 Å. The dose rate was 2.6–2.8 e-/$Å^2$/s during 9-s exposures, resulting in the total dose of 23–25 e-/$Å^2$ on the specimen. These exposures were collected manually and fractionated into 36 movie frames (0.25 s/frame) with SerialEM (http://bio3d.colorado.edu/SerialEM/), at defocus ranging from −0.4 to −2.5 µm defocus.

## Cryo-EM image processing and 3D reconstruction

Movie frames were aligned using MotionCor2 (Zheng *et al*, 2017), with images divided to 25 tiles for local correction of beam-induced motion. Start and end positions for each uninterrupted MT were manually selected using the helix mode in EMAN 1 Boxer (Ludtke *et al*, 1999) that automatically generated 652 × 652 pixel boxes with 474 pixel overlap along MTs. This box size corresponded to MT segments spanning ~11 tubulin dimers with ~8 dimer overlap between the segments.

For 3D reconstruction of 13-PF MTs, the MT segments boxed with EMAN1 were averaged in Spider (Frank *et al*, 1996). Segment averages revealing: i) MT architecture other than 13-PF (in the case of NN-MTs), ii) lattice defects, and/or iii) poor contrast (e.g., blurring) were excluded from further processing. The selected high-quality 13-PF segments were treated as single particles in Chuff (Sindelar & Downing, 2007), a custom-designed multi-script processing pipeline using Spider (Frank *et al*, 1996) and Frealign (Grigorieff, 2007). The initial 2D alignment was aimed at identifying the MT seam location by projection matching in Spider. The reference projections for this step were generated using a synthetic 13-PF DCX-MT reference volume filtered to 30 Å. The contrast transfer

function (CTF) parameters for each micrograph were estimated with CTFFIND3 (Mindell & Grigorieff, 2003), and the CTF correction was performed during local refinement within Frealign, producing isotropic 3D reconstructions with pseudo-helical symmetry applied 12 times. This symmetry operation was chosen to average only the PFs occupied with DCX, excluding the unoccupied seam. Independently processed half maps were combined and sharpened in Relion 1.4 (Scheres, 2012) through its automated post-processing routine. The average resolutions of the final maps were estimated using 0.143 FSC criterion and the absence of over-fitting was confirmed with high-resolution noise substitution test (Chen *et al*, 2013; Table 1, $FSC_{true}$).

For 3D classification and refinement of 13- and 14-PF NN-MTs, the MT start and end coordinates selected using EMAN 1 (as above) were imported to Relion 2.1 (Kimanius *et al*, 2016), and 432 × 432 pixel MT segments spaced with 1 dimer distance were generated. 3D classification was done with 15 $Å^2$ resolution cut-off using 4× binned segments (resulting in 5.56 $Å^2$ pixel size) against six synthetic volumes representing 11-, 12-, 13-, 14-, 15-, and 16-PF MT architectures. The most populated 13-PF and 14-PF classes (together containing 99% of all MT segments) were then separately refined, applying pseudo-helical symmetry 12 times (for 13-PF class) or 13 times (for 14-PF class). The resultant 3D maps were sharpened according to local resolution and the average resolution was estimated as before.

### Cryo-EM data analysis

In the single-particle, cryo-EM data processing so-called "bad particles" are excluded due to their obvious poor quality, which precludes their alignment with the consensus data. The sparse regions of micrographs where sample image quality is poor (for example, due to grid surface contamination giving rise to local noise in the image or due to sample heterogeneity) would ideally be not selected for processing, but it is impossible to completely avoid it, even with manual particle/segment picking used in this study. Image processing algorithms employed reveal such poor particles as not classifiable under objective computational criteria into any biologically relevant class, which objectifies exclusion.

### Determination of NDC occupancy on MTs

Fourier transforms of the relevant MT segments were averaged using EMAN 1 (Ludtke *et al*, 1999). Intensities of 8 nm layer lines corresponding to MT-binding periodicity of NDC in relation to 4 nm layer lines corresponding to tubulin subunit periodicity were then calculated using FiJi (https://fiji.sc/). To estimate the error of these ratios we divided each dataset into three approximately equal subsets and calculated the average ratio and standard deviation within each set of three subsets.

### Atomic model refinement

The 1MJD NMR structure of human NDC (Kim *et al*, 2003) or the 5IP4 crystal structure of human CDC (Burger *et al*, 2016) were combined with previously refined tubulin subunits (Manka & Moores, 2018) to create starting models for MT-bound DC domain refinements in our cryo-EM maps. The maps were zoned in UCSF

Chimera (Pettersen *et al*, 2004) around DC domains surrounded by 4 (2 β and 2 α) tubulin subunits. Each isolated map was placed in a new unit cell with P1 space group. The model-map fit was adjusted in Coot (Emsley *et al*, 2010) before 10 macro cycles of refinements in real space were carried out using phenix.real_space_refine (http://phenix-online.org/; Afonine *et al*, 2018) with default settings. Phenix automatically determined weight between data and the restraints to achieve RMS deviations of covalent bonds not greater than 0.01, and for angles not greater than 1.0 from ideal values. Manual adjustments of poorly fitting regions in Coot followed by real space refinements described above were repeated until a satisfactory level of model: map agreement and excellent model geometry were accomplished (Table 1).

### Structure analyses and presentation

Analyzes and visualizations of our cryo-EM density maps and the atomic models refined in those maps were done using PDBePISA (http://www.ebi.ac.uk/msd-srv/prot_int/cgi-bin/piserver) (Krissinel & Henrick, 2007), PyMOL (Schrödinger), UCSF Chimera (Pettersen *et al*, 2004), and ChimeraX (Goddard *et al*, 2018).

## Data availability

Our cryo-EM density maps and atomic coordinates were deposited in the Electron Microscopy Data Bank (EMDB) and the Protein Data Bank (PDB) with the following accession codes: 13-PF WT(CDC)-GTP/GDP.Pi-MT (EMD-4861; PDB 6RF2), 13-PF WT(NDC)-GDP-MT (EMD-4858; PDB 6REV), 13-PF NN(NDC)-GDP-MT (EMD-4862; PDB 6RF8), 14-PF NN(NDC)-GDP-MT (EMD-4863; PDB 6RFD).

*Expanded View* for this article is available online.

### Acknowledgements
This work was funded by grants from the Medical Research Council, U.K. to C.A.M (MR/R000352/1). EM data collection was supported by grants from the Wellcome Trust (079605/Z/06/Z, 101488/Z/13/Z) and the BBSRC (BB/L014211/1). Anthony Roberts provided essential advice concerning TIRF microscopy while Joe Atherton and Alex Cook provided valuable guidance about MT processing using RELION.

### Author contributions
SWM conceived experimental strategies, designed and carried out experiments and computations, analyzed data, interpreted results, and wrote the manuscript; CAM proposed and supervised the research, interpreted results, and wrote the manuscript.

### Conflict of interest
The authors declare that they have no conflict of interest.

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
