## [Review Process File · EMBO Reports]

Pseudo-repeats in doublecortin make distinct mechanistic contributions to microtubule regulation

Szymon Manka and Carolyn Moores
DOI: [10.15252/embr.202051534](https://doi.org/10.15252/embr.202051534)

Corresponding author(s): Carolyn Moores (c.moores@mail.cryst.bbk.ac.uk) , Szymon Manka (s.manka@mail.cryst.bbk.ac.uk)

Review Timeline:	Submission Date:	17th Aug 20
	Editorial Decision:	8th Sep 20
	Revision Received:	17th Sep 20
	Accepted:	18th Sep 20

Transaction Report: This manuscript was transferred to EMBO reports following peer review at The EMBO Journal.

Institute of Structural and Molecular Biology,
Birkbeck, University of London,
London WC1E 7HX,
UK

17/08/20

EMBO Reports

Dear Dr Rembold

Thank you very much for considering the revised version of our manuscript entitled **“Pseudo-repeats in doublecortin make distinct mechanistic contributions to microtubule regulation”**, transferred from EMBO Journal (EMBOJ-2019-102526R), for consideration at *EMBO Reports*.

We have addressed the comments of the referees, described in detail in the point by point response. In summary, however, given the already substantial amount of data included in our manuscript, and in particular our very high quality cryo-EM reconstructions, we considered additional experiments to be beyond the scope of our current study. Therefore, at the request of Referee 1, we have removed the Rosetta- and single molecule TIRF-related material. Together with some reorganisation of the Results section, this has enabled a clearer presentation of our other data, which constitutes the majority of the work.

We are not aware of published data that impacts the novelty of our work. However we know that our work is being read and found relevant by the community, and that our findings (as posted on bioRxiv; <https://doi.org/10.1101/808899>) have been used and cited by others in related work (Rogers et al, <https://doi.org/10.1101/2020.06.12.149252>). We have also included a citation of this recent preprint about the DCX relative DCX-like kinase that is relevant to, and consistent with, our study of DCX.

We have highlighted all these text changes, together with those from the previous revision at EMBO Journal, in the submitted manuscript file.

We believe that this study would be very suitable for publication at EMBO Reports. Thank you once again for your consideration and we look forward to hearing from you in due course.

Yours sincerely

Carolyn Moores and Szymon Manka

Point-by-point response

Referee #1:

The revised Manka et al manuscript, presented written rebuttals to the three reviewer comments. The reviewers appear to be in agreement regarding concerns about the simulated binding of the CDC domain to tubulin protofilaments. However, the revised manuscript provides no additional data to address the biochemical relevance of the authors hypothesis regarding differences between the CDC and the NDC binding MT ends using single molecule methods or biochemical data regarding CDC binding curved protofilaments in solution. This reviewer believes that biochemical data for the CDC binding to curved protofilaments (#1), and quantitative analyses on ensembles of single molecule data (#2) are necessary for these conclusions to be included in the manuscript. These data are not interpretable only in the manner the authors suggest. However, the authors did not provide any new data or analyses to address these concerns. The other reviewers agree with these points and suggest that the manuscript has focused on these areas which take away from the main message for the manuscript.

1) The CDC binding to curved tubulin protofilaments was only simulated using Rosetta docking. Although the CDC-tubulin protofilament docking is a reasonable place to start a for a hypothesis, it remains no more than a hypothesis. From the understanding of this reviewer that this hypothesis invokes a binding site for CDC between the corners of four curved tubulins in two protofilament pairs and not single protofilaments. The authors show images of protofilaments in their EM imaging but provide no evidence that CDC is bound to them. This could have been easily accomplished using has been no biochemical data has been presented. The authors also assert that the clinical mutations provide proof for this hypothesis. The mechanism of these mutations remains unknown without direct testing with biochemical data. The authors argument in the rebuttal appears a bit logically "circular" in that they identify a new conformation of the CDC (using Rosetta) in which residues make new types of contacts, then the authors asset that conservation of these residues supports the importance of this conformation. Without having a biochemical measure, these data remain a highly unproven hypothesis and should be removed.

2)The single molecule studies differentiating the binding and dissociation parameters for the wt and NN DCX construct remain a major concern due to the lack of quantitatively studies on larger ensembles. Furthermore, the images of kymographs lack clear images, steady signal and seem to be missing quantitative measures to establish how these molecules bind and dissociate differently from MT plus-ends. The authors state the data presented is "quality is sufficient for its mechanistic purpose". This reviewer completely disagrees with this point and remains unconvinced by these data. The lack of statistical analyses for the residence times of moderately sized ensembles of single molecule data in the experiments presented leaves a major concern about how many data points were used to make the conclusions presented by the authors.

The data associated with both the above sections should either be removed from the manuscript or fortified with additional data. Currently, these data detract away from the main story which remains of high quality.

1. Given the already substantial amount of data included in our manuscript, and in particular our very high quality cryo-EM reconstructions (as also endorsed by this referee), we considered additional experiments to be beyond the scope of our current study. However, in line with the suggestions of this referee, together with referee 3, we have removed the Rosetta and single molecule TIRF data. We believe the consequent reorganisation of the text allows a much clearer description of the remaining majority of our data. Our figures/data are now organised as follows:

Figure 1 -> remains Figure 1, with added information about statistical significance in panel I
Figure 2A-D -> new Figure 2, which also contains new views of the MT binding surfaces
Figure 2E-G -> moved to new Figure 4
Figure 3 -> remains Figure 3
Figure 4 parts relating to Rosetta work -> removed;
Figure 4 parts relating to comparison of our structures -> moved to new Figure 5
New Figure 4 – detailed view of CDC and NDC MT binding sites
Figure 5 (single molecule TIRF data) -> removed
New Figure 5 – further analysis of CDC and NDC structures and MT binding sites
Figure 6 -> remains Figure 6
Figure 7 -> remains Figure 7, with Rosetta-related features removed
Table 1 -> remains Table 1

Figure EV1 -> remains Figure EV1
Figure EV2 -> remains Figure EV2
Figure EV3A (relates to cryo-EM data) -> moved to new Figure 5B
Figure EV3B, C (top row) and E (relating to Rosetta work) -> removed
Figure EV3C (bottom rows) and D -> rearranged to form a new Figure EV3
Figure EV4 -> remains Figure EV4
Figure EV5 -> remains Figure EV5

Appendix Figures S1, S2, S3, S4 -> remain unchanged
Appendix Figure S5 -> expanded to local resolution depictions and the low-pass filtered version of our reconstructions, mentioned by referee 2 (below)
Appendix Figure S6 -> remains unchanged
Appendix Figure S7 relating to Rosetta work -> removed;
Appendix Figure S8 relating to single molecule TIRF data -> removed

Appendix Table S1 -> remains Appendix Table S1

Referee #2:

The response (especially the figure included in the response file) adequately addressed my main concerns regarding the structure data interpretation. I am happy to accept this manuscript for publication. Congratulations to the authors for an interesting and detailed study.

2. Thanks to this referee for their support of our work. We note that the mentioned figure that was originally included in the author response file, which shows a low resolution filtered view of our reconstruction, is now included in Appendix Figure S5C.

Referee #3:

The authors clearly have made an effort to address the reviewer's comments and I generally support publication. This is still a very dense paper and I don't think completely resolves how DCX interacts with microtubules, but this is an important step in this direction.

3. Removal of the Rosetta docking data and the single molecule TIRF data has allowed us to unpack the rest of our results, which we hope helps to make our manuscript less dense.

There are still a few things that do not make sense to me:

Point 27:

[Previous discussion for reference:

Fig 1D: The fact that the NN construct is a less efficient MT nucleator compared with WT does not necessarily imply that the C-terminal DCX domain has that function; just means that the NN construct works less well than WT, which could be for a number of reasons. This becomes important when moving to Fig. 2 when it is implied that the different DCX domains have different functions that can be clearly separated. Yet, no function of the CDC domain has been shown (because of its purported instability; how does one distinguish stability of the CDC domain from its MT affinity? Could the two be related?). If it is indeed the case that CDC promotes MT nucleation before NDC binds, why would the C-terminal domain constructs alone not promote nucleation then (i.e. Fig 1C). maybe I am missing something, but this seems confusing.

27. We understand the reviewer's concerns which arise partly from the deficit of our original presentation as also highlighted by reviewer 2 (see our response in particular in point 12 and 20). Our data and the rest of the literature are very clear that 2 DC domains are required for full DCX functionality. Our data now show that the identity of these 2 domains are important and provide a model for their distinct roles. With the addition of the gel illustrating the purified proteins in our study (Appendix Figure S3) we now show that protein stability is a major issue for the CC and NC constructs. However, in the NN construct we show not only that the absence of CDC slows down MT nucleation, but also that it causes heterogeneity in MT architecture. We think this is a strong evidence for the role of CDC in efficient nucleation of physiological 13-protofilament MTs. We now also show in Fig EV1F that our NN construct induces faster MT growth from pre-existing MT seeds (which essentially bypasses MT nucleation step) than the WT. Despite this ability to accelerate MT polymerization, NN is a less potent in inducing formation of MT seeds (MT nucleation), which strengthens our conclusion about the importance of CDC for MT nucleation.

We have now improved the explanation of the proposed mechanism of DCX action to emphasise the importance of the two-domain binding (Abstract: lines 18-22; Introduction: lines 90-91; Results: lines 175-180 and 189-192; Discussion: lines 362-375 and 404-419).]

Nearly all microtubules in mammalian cells are thought to have 13 protofilaments. Yet, only developing neurons express DCX. So, I do not quite understand how the function of the CC domain can be to nucleate 13 protofilament microtubules. Aren't they nucleated anyways? Maybe a sentence or two explaining this would be good.

4. As we mention at the beginning of the Discussion text, DCX "localises to the distal processes of the migrating immature neurons (Tint et al., 2009), where the absence of the centrosomal γ -tubulin ring complexes (γ -TuRCs) that normally template the 13-PF architecture, may require that DCX contributes to *de novo* [13-PF] MT nucleation to help direct cell migration (Stiess et al., 2010). " We have now added the text in brackets to the latest version of our manuscript, to further emphasise the hypothesised role of DCX in specifying the protofilament architecture of *de novo* nucleated MTs in migrating neurons.

Point 31:

Overall, this experiment does not show that one domain is replaced by the other, only that CDC appears to have an increased affinity for GDP.Pi, while NDC likes GDP better.

31. We agree that it directly shows the apparent affinity difference, but we also show that CDC binds first - to the GDP.Pi lattice - and the subsequent GDP lattice is dominated by NDC, thus one domain must be replaced by the other during the MT growth in the conditions of the reaction (large DCX excess over the binding sites on MTs).

ok, but it does not necessarily show that a domain replacement happens in the same way at a growing MT end where the GTP-lattice state would turn over much more rapidly than 1 hour. I think this experimental caveat should be pointed out somewhere (maybe at the end of the relevant paragraph; top of page 5).

5. This is a fair point. Edits relating to this point are incorporated as part of the wider changes in the text, in particular on p6.

Point 37:

[Previous discussion for reference:

Fig 5: I understand the attractiveness of the model that CDC binds curved protofilaments at growing MT ends and NDC binds the GDP lattice. However, I don't think the experiments in Fig. 5 clearly demonstrate this. If CDC specifically recognized longitudinally curved protofilaments at growing MT ends, I would expect tracking of CDC with the growing end similar to what has been shown by others with DCX protein in vitro. It does not make a lot of sense to me that CDC would become incorporated at a specific position on the MT and stay there (presumably once this part of the MT is no longer the end, protofilaments would be no longer curved. Is the idea that CDC lets go and NDC binds instead? However, this is all conjecture and why would NN with two high affinity NDC domains not kick off the WT construct with only one NDC especially at a 50-fold excess). Have the authors tested if any of the single CDC domain constructs track MT ends?

37. We thank the reviewer for pointing this out. The illustration in Fig 5 that explains the observed behaviour of WT-SNAP and NN-SNAP (panels H and I) has now been improved. Despite the 50-fold excess of NN over WT-SNAP in the competition experiment, the total DCX concentration is still ~10 fold lower in relation to tubulin, as the reviewer rightly pointed out earlier, so it is very likely that NDC joins CDC in binding to the straight MT lattice in these competition experiments. The NN excess is not super-stoichiometric in relation to tubulin and it can be easily reconciled with WT-SNAP binding, even with the 50-fold excess of NN. We have not tested the single CDC constructs since these were not carried forward in the course of the study, when they did not show any of the physiological DCX functions.

We speculate in the Discussion that the observed lack of tip tracking is due to isoform differences in the various studies (lines 448-466). Both tip tracking and lack of tip tracking have been reported in different experiments with different DCX isoforms; the cooperative DCX binding phenomenon that can explain DCX retention at its site of association with MT has also been reported, but its mechanism is also not known. We propose in the Discussion that the cooperative binding that effectively results in locking of DCX molecules in place by the neighbouring DCX molecules may rely on MT lattice regularization by DCX (lines 411-419). This regularization helps to achieve high resolution in cryo-EM.]

I don't know, but I find it hard to believe that a 5 aa difference distinguishes between tip tracking and lattice binding. It is also maybe worth mentioning that the Ettinger paper showed that in cells WT DCX not only does not track MT ends, but is absent from growing microtubule ends. So, this remains incompatible with the model in Fig. 7 in which DCX through its CDC only binds to the GDP-Pi zone. Granted, cells and in vitro are different, but this is still odd.

6. These comments originated from a discussion of the single molecule TIRF data, which have now been removed. Although no specific questions are put to us, we nevertheless note that the referee raises two additional points:

1) Small isoform-specific differences may not be sufficient to explain observations of tip tracking vs lattice binding by DCX.

-> In the Discussion text, we simply offer isoform-specific differences as one potential explanation for these observations that cannot currently be excluded. Future studies building on our current work will hopefully reveal more about isoform-specific behaviours of DCX.

2) That our model is incompatible with the observations of Ettinger et al, where WT DCX not only does not track MT ends, but is absent from growing microtubule ends in cell.

> Our model focuses on DCX's involvement in de novo microtubule nucleation, which we speculate is important in neurons and where reported local concentrations of DCX support the feasibility of this model (Tint et al, 2009). This is described in our Discussion text. While we appreciate the work of Ettinger very much and cite it in several places, our reading of their work is that they did not explicitly investigate de novo nucleation, nor do they evaluate which other factors might be influencing DCX's occupancy of the various microtubule regions in the cells of their study. Nevertheless, their observation that DCX favourably binds regions of curved MTs induced by paclitaxel addition – which have been proposed to have some structural similarities to curved, early microtubule nuclei and/or microtubule tips - are consistent with our model as we already state in our Discussion text, and also hint at a more complicated situation in cells. Our data concerning the specific roles of NDC and CDC within DCX provide unique insights that can inform future investigations that will illuminate these topics further.

Dear Prof. Moores

Thank you for the submission of your revised manuscript to EMBO reports. Your study had been reviewed and revised for our sister journal The EMBO Journal. While reviewer 2 was satisfied with the revised manuscript, reviewer 1 and 3 found that some of the data remained inconclusive and considered a more thorough biochemical characterization of the Rosetta and single molecule data essential. We had offered publication of a further revised version in EMBO reports given that the conclusions derived from the molecular docking simulations and single molecule experiment data would be appropriately toned down and the limitations clearly discussed or the respective data be removed.

You have now submitted such a revised version and after checking all the files and your point-by-point response, I am happy to offer publication after some remaining editorial concerns have been addressed as follows:

- Data availability section: please provide links that resolve to the deposited datasets
- Appendix table of content: please add page numbers
- There are callouts to Supplementary Table 1 in the legend of Figure 6. Please correct the callouts to "Appendix Table S1".
- You mention that you analyzed movie files and we note that movies were part of the earlier submission to our sister journal The EMBO Journal. Please ensure that all relevant files have been uploaded.
- Please change the header of "Declaration of Interests" to "Conflict of Interest"
- Please add callouts to Fig 7A and Fig EV 1F to the text.
- Please provide a complete author checklist, which you can download from our author guidelines (<<https://www.embopress.org/page/journal/14693178/authorguide>>). Please insert information in the checklist that is also reflected in the manuscript. The completed author checklist will also be part of the Review Process File.
- During our routine image analysis, we noted that some figures were too closely cropped, i.e., there was little space on the bottom/right edge of the figure, which might create problems when published online. We have now added a white border (Figures 1,3,5,7 & Figures EV1, EV2, EV5).
- I attach to this email a related manuscript file with comments by our data editors. Please address all comments and upload a revised file with tracked changes with your final manuscript submission.
- Finally, could you please add a draft for a short summary text to the synopsis file I attached (1-2 sentences)?

With kind regards,

Martina Rembold, PhD
Editor
EMBO reports

The authors have addressed all minor editorial requests.

Prof. Carolyn Moores
Institute of Structural and Molecular Biology, Birkbeck College
Department of Biological Sciences
Malet Street
London WC1E 7HX
United Kingdom

Dear Prof. Moores,

I am very pleased to accept your manuscript for publication in the next available issue of EMBO reports. Thank you for your contribution to our journal.

At the end of this email I include important information about how to proceed. Please ensure that you take the time to read the information and complete and return the necessary forms to allow us to publish your manuscript as quickly as possible.

As part of the EMBO publication's Transparent Editorial Process, EMBO reports publishes online a Review Process File to accompany accepted manuscripts. As you are aware, this File will be published in conjunction with your paper and will include the referee reports, your point-by-point response and all pertinent correspondence relating to the manuscript.

If you do NOT want this File to be published, please inform the editorial office within 2 days, if you have not done so already, otherwise the File will be published by default [contact: emboreports@embo.org]. If you do opt out, the Review Process File link will point to the following statement: "No Review Process File is available with this article, as the authors have chosen not to make the review process public in this case."

Should you be planning a Press Release on your article, please get in contact with emboreports@wiley.com as early as possible, in order to coordinate publication and release dates.

Thank you again for your contribution to EMBO reports and congratulations on a successful publication. Please consider us again in the future for your most exciting work.

Yours sincerely,

Martina Rembold, PhD
Editor
EMBO reports

THINGS TO DO NOW:

You will receive proofs by e-mail approximately 2-3 weeks after all relevant files have been sent to our Production Office; you should return your corrections within 2 days of receiving the proofs.

Please inform us if there is likely to be any difficulty in reaching you at the above address at that time. Failure to meet our deadlines may result in a delay of publication, or publication without your corrections.

All further communications concerning your paper should quote reference number EMBOR-2020-51534V2 and be addressed to emboreports@wiley.com.

Should you be planning a Press Release on your article, please get in contact with emboreports@wiley.com as early as possible, in order to coordinate publication and release dates.

YOU MUST COMPLETE ALL CELLS WITH A PINK BACKGROUND ↓
PLEASE NOTE THAT THIS CHECKLIST WILL BE PUBLISHED ALONGSIDE YOUR PAPER

Corresponding Author Names: Szymon W. Manka and Carolyn A. Moores

Journal Submitted to: EMBO reports

Manuscript Number: EMBOR-2020-51534V1